# Structured Unrestricted-Rank Matrices for Parameter Efficient Fine-tuning

**Arijit Sehanobish**[1*] **Avinava Dubey**[2*] **Krzysztof Choromanski**[3,4*]
**Somnath Basu Roy Chowdhury**[5*] **Deepali Jain**[3] **Vikas Sindhwani**[3] **Snigdha Chaturvedi**[5]

[1]Independent   [2] Google Research   [3]Google DeepMind
[4]Columbia University   [5]UNC Chapel Hill

## Abstract

Recent efforts to scale Transformer models have been successful across a wide range of tasks [77]. However, fine-tuning these models for downstream tasks can be expensive, as it requires updating a large number of parameters in the Transformer model. Parameter-efficient fine-tuning (PEFT) approaches have emerged as a viable alternative that allow us to fine-tune models by updating only a small number of parameters. In this work, we propose a general framework for parameter efficient fine-tuning using *structured unrestricted-rank matrices* (SURM), which can serve as a drop-in replacement for popular approaches such as Adapters and LoRA. Unlike other methods like LoRA, SURMs provides more flexibility in finding the right balance between compactness and expressiveness. This is achieved by using *low displacement rank matrices* (LDRMs), which has not been used in this context before. SURMs remain competitive with baselines, often providing significant quality improvements while using a smaller parameter budget. SURMs achieve **5**-**7**% accuracy gains on various image classification tasks while replacing low-rank matrices in LoRA. It also results in up to **12x** reduction of the number of parameters in adapters (with virtually no loss in quality) on the GLUE benchmark.

## 1   Introduction

In recent years, large-scale Transformer models have demonstrated impressive performance across a wide range of domains, including natural language processing (NLP) [20, 8], vision [36], robotics [7], and even multi-modal settings [81]. For many applications, a single large pre-trained model is *adapted* for several downstream problems. *Fine-tuning*, where all the model parameters are updated, is a popular way to adapt a pre-trained model to a new task or domain. However, fine-tuning large models on specific downstream tasks requires significant computational resources and involves a massive memory footprint, as each task necessitates storing its own set of parameters.

Parameter-efficient fine-tuning (PEFT) methods have emerged as the preferred methodology to adapt pre-trained Transformers to different downstream tasks. PEFT methods often achieve performance on par with full fine-tuning while training only a small number of parameters [80, 45]. PEFT techniques involve either training a small subset of the model's parameters [84, 42] or integrating small modular layers while freezing the base model's weights [26, 25]. There are two popular classes of methods to inject additional parameters: **(a)** using small modular layers inside Transformers called *adapter* layers [59], and **(b)** constraining the updates as *low-rank matrices* (**LoRA**) [26].

Although adapters and LoRA (including their variants) differ architecturally and conceptually, they share a common reliance on low-rank matrices. The success of these methods has been attributed to the low intrinsic dimensionality of the hidden representations in the pre-trained Transformer

---

[*]Equal Contribution

38th Conference on Neural Information Processing Systems (NeurIPS 2024).

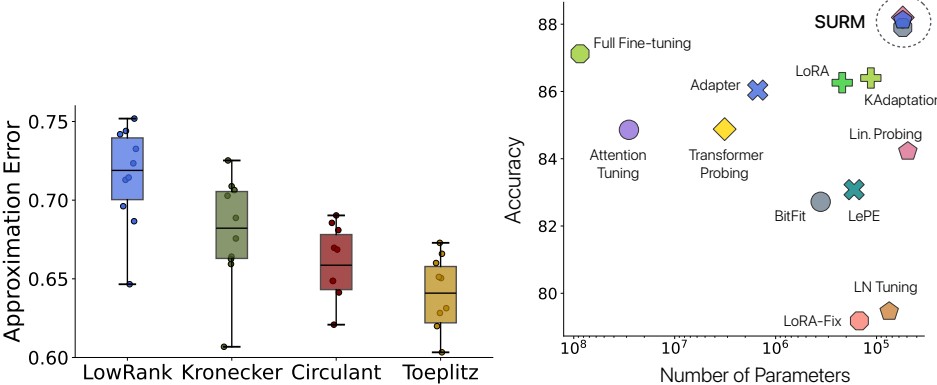

Figure 1: **Left:** Approximating a PSD matrix using a low rank matrix, Kronecker product of matrices, circulant matrix, and Toeplitz matrix. We repeat our experiment **10** times and for each trial, we observe that low rank matrix is the worst approximator followed by Kronecker product, circulant, and Toeplitz. **Right:** The tradeoff between accuracy and parameter numbers of various PEFT methods. Results are measured across 5 image datasets using CLIP-ViT. Our methods appear in the top right corner (in blue) and achieve the best performance among various strong baseline methods.

models [1, 70]. These low-rank methods primarily aim to approximate updates, which, in general, are not low rank. Hence, there's no justification for imposing low-rank constraints on them. Motivated by this insight, we explore alternative classes of matrices—ones that aren't necessarily low rank but are characterized by a linear number of parameters while exhibiting impressive approximations across various matrix classes. We present Fig. 1 as a preview of the motivating results. In Fig. 1 (left), we show that structured matrices (SURM) can approximate any random matrix better than low rank matrices. In Fig. 1 (right), we show that when SURMs are used for parameter efficient fine-tuning it outperforms existing PEFT methods (see more details in Sec. 4).

We propose a novel paradigm of parameter efficient fine-tuning that leverages *Structured Unrestricted-Rank* matrices (or SURMs). SURMs provide similar efficiency gains as previous works in efficient fine-tuning, but their more expressive structure paves the way for quality improvements. In this work, we propose to perform parameter-efficient fine-tuning by parameterizing learnable weights as structured matrices. We focus on the two sub-classes of SURMs: **(1)** Kronecker product of matrices [3] and **(2)** low displacement rank matrices (LDRMs) [66, 6, 58, 54, 73]. To summarize, our primary contributions are:

- We propose the class of Structured Unrestricted-Rank matrices (SURMs) (Section 3), for parameter efficient fine-tuning of Transformers. SURMs include low-rank matrices used in LoRA, as special cases. To the best of our knowledge, we are the first to apply LDRMs in this context.

- We demonstrate strong matrix approximation capabilities inherent in Low Displacement Rank Matrices, with a specific focus on circulant and Toeplitz matrices (Section 4).

- We introduce a new class of adapter-layers using SURMs, achieving a **12x** reduction in parameters compared to adapters, with virtually no loss in quality on the GLUE benchmark (Section 6).

- We achieve **5-7**% accuracy gains over LoRA on a wide variety of image datasets as well as in low resource setting (VTAB-1k benchmark). In some cases SURMs outperform full fine-tuning, while using only **55**k training parameters (as shown in Fig. 1 (right)).

## 2  Related Work

With the introduction of BERT [20] and GPT-2 [8], Transformer models trained on general text corpora have revolutionized the field of machine learning (ML). Since then, these models have continued to increase in size, with open-source variants adopting various architectures. Examples include encoder-decoder models such as T5 [64] with up to 20B parameters [67], and a range of auto-regressive decoder models like Llama [69], Pythia [4], Mistral [28], among others, varying in size from a few million to 180B parameters [2]. These models can be easily adapted to downstream tasks by fine-tuning on task-specific data, resulting in state-of-the-art performance across a broad

spectrum of downstream tasks. Due to the computational infeasibility of fine-tuning all the parameters of these models, in-context learning [8] and prompt engineering [11, 22] have emerged as attractive alternatives to adapt models to downstream tasks. However, such adaptation results depend heavily on the design of the input prompt and tend to vary greatly with small perturbations of the prompts [48].

Consequently, many works have proposed various PEFT techniques. One of the earliest methods involves inserting the so-called *adapter* layers between existing layers in a neural network [25, 59]. An adapter is typically an MLP with input, output, and a smaller middle layer, encoded by two low-rank matrices, making it compact in terms of parameters. An extension of the adapter is Compacter [51], which uses Tucker decomposition to parameterize the adapter layers and weight-sharing to reduce the number of trainable parameters. Various modifications and extensions of the above methods have been proposed [53, 24, 32, 52, 65]. Another popular PEFT technique is differentiable prompt-tuning (DPT), which can be thought of as optimizing special tokens in the prompt [88]. However, these methods are limited by the sequence length of the underlying models. Even though DPT was originally developed for NLP, several works have extended it for computer vision tasks as well [83, 12, 27, 24].

One of the most popular PEFT methods is Low-Rank Adaptation (LoRA) [26], which imposes a low-rank constraint on the weight updates. The main difference between adapters and LoRA is that the learned LoRA weights can be merged with the frozen model weights during inference without adding any latency. Given the popularity of LoRA, there have been many works on extending it to different contexts like long-range modeling [13], multi-tasking [10] or improving its efficiency [19, 71, 46, 37, 31] among many others.

In general, low-rank matrices are studied extensively in various ML applications [57, 87, 44, 61]. The research on low displacement rank matrices (LDRMs) for ML is more narrow [89, 68, 41, 66, 15, 35, 62]. Although Kronecker matrices (a class of LDRMs) have been explored in the context of LDRMs [21, 24, 51], the constituent matrices in the Kronecker product have low rank even in these work. In this work, we use a fixed parameter budget but do not impose any rank-based condition. To the best of our knowledge, we are the first to systematically explore the effectiveness of different structured matrices and introduce LDRMs for parameter-efficient fine-tuning.

The rest of the paper is organized as follows: **(a)** We introduce the notion of Structured Unrestricted-Rank Matrices (SURM) that are used in this work (Section 3), **(b)** We motivate the usage of SURM by empirically showing the approximation qualities of these matrices (Section 4), **(c)** We use SURM as drop-in replacement for popular approaches such as Adapters and LoRA (Section 5), **(d)** We validate our approach across a wide range of vision and NLP tasks (Section 6).

## 3 Structured Unrestricted-Rank Matrices (SURM)

In this section, we will define the matrices that are used for parameter efficient fine-tuning. First, we define the concept of a *structured matrix*, which is a generic term for a matrix $\mathbf{A} \in \mathbb{R}^{m \times n}$ that can be represented by fewer than $mn$ parameters. These matrices are useful because they reduce both space and time complexity when performing matrix multiplications.

A simple example of a structured matrix is a low rank matrix of the form $\mathbf{W} = \mathbf{A}\mathbf{B}^\top \in \mathbb{R}^{m \times n}$, where $\mathbf{A} \in \mathbb{R}^{m \times r}$, $\mathbf{B} \in \mathbb{R}^{n \times r}$ with $r \ll \min(m, n)$. In this work, our main focus is on those classes of structured matrices that are not restricted to be low-rank, which we refer to as *Structured Unrestricted-Rank Matrices* (SURM). Next, we present two classes of SURM matrices that we use for parameter efficient fine-tuning.

**Low Displacement Rank Matrices**. Our first class of SURMs is low displacement rank matrices (LDRMs). A matrix $\mathbf{W} \in \mathbb{C}^{m \times n}$ is said to have $(\mathbf{A}, \mathbf{B})$-displacement structure if:

$$\nabla_{\mathbf{A},\mathbf{B}}(\mathbf{W}) \overset{\text{def}}{=} \mathbf{A}\mathbf{W} - \mathbf{W}\mathbf{B} = \mathbf{F}, \tag{1}$$

where $\mathbf{A} \in \mathbb{C}^{m \times m}, \mathbf{B} \in \mathbb{C}^{n \times n}, \mathbf{F} \in \mathbb{C}^{m \times n}$ and $\mathbf{F}$ has low rank $r$ (as compared to $\min(m, n)$). We call $\nabla_{\mathbf{A},\mathbf{B}}$ the *displacement rank operator*, parameterized by $\mathbf{A}$ and $\mathbf{B}$.

For a given $\mathbf{W}$, there can exist several pairs of $(\mathbf{A}, \mathbf{B})$ matrices satisfying Eq. 1 that produce a low-rank matrix, $\mathbf{F}$. Some examples of such $(\mathbf{A}, \mathbf{B})$ pairs include: $(\mathbf{Z}, \mathbf{Z}), (\mathbf{Z}, \mathbf{Z}^\top), (\mathbf{D}_x, \mathbf{Z}^\top), (\mathbf{D}_x, \mathbf{D}_y)$ (for $x \neq y$). Here $\mathbf{Z}$ is a circulant-shift matrix and $\mathbf{D}_z$ is a diagonal matrix with nonzero entries equal to $z$. Low displacement rank matrices ($\mathbf{W}$ in Eq. 1) enable fast (sub-quadratic) matrix-vector multiplication and enhance the efficiency of other matrix operations, such as inversion. By selecting

$$
\begin{bmatrix}
c_0 & c_{n-1} & \cdots & c_2 & c_1 \\
c_1 & c_0 & c_{n-1} & \cdots & c_2 \\
\vdots & c_1 & c_0 & \ddots & \vdots \\
c_{m-2} & \vdots & \ddots & \ddots & \vdots \\
c_{m-1} & c_{m-2} & \cdots & \cdots & c_m
\end{bmatrix}
\qquad
\begin{bmatrix}
a_0 & a_{-1} & \cdots & \cdots & a_{-(n-1)} \\
a_1 & a_0 & a_{-1} & \cdots & \vdots \\
a_2 & a_1 & a_0 & \ddots & \vdots \\
\vdots & \vdots & \ddots & \ddots & a_{-1} \\
a_{m-1} & \cdots & \cdots & a_1 & a_0
\end{bmatrix}
\qquad
\begin{bmatrix}
a_{11}\mathbf{B} & \cdots & a_{1n}\mathbf{B} \\
\vdots & \ddots & \vdots \\
a_{m1}\mathbf{B} & \cdots & a_{mn}\mathbf{B}
\end{bmatrix}
$$

|  (a) Circulant | (b) Toeplitz | (c) Kronecker |
| --- | --- | --- |

Figure 2: A schematic diagram to illustrate the structure (a) Circulant, (b) Toeplitz, and (c) Kronecker product of two matrices $\mathbf{A}$ and $\mathbf{B}$.

more complex $(\mathbf{A}, \mathbf{B})$ pairs, such as those involving *general Jordan form matrices*, it is possible to consider more unstructured $\mathbf{W}$ that still have compact representations and support efficient matrix operations [56, 66]. In this paper, we focus on classic low displacement rank matrices: circulant and toeplitz matrices, which are described below.

1. **Circulant Matrices**: A circulant matrix $\mathbf{C} \in \mathbb{C}^{m \times n}$ can be parameterized by its first row. The following rows are obtained from the previous one by applying a right circulant shift. A schematic visualization of a circulant matrix is shown in Fig 2 **(a)**. Since we only need to store the first row, circulant matrices can be trivially encoded in $O(n)$ space. They also support fast matrix-vector multiplication in $O((n + m) \log(n + m))$ time using Fast Fourier Transform (FFT) [55].

2. **Toeplitz Matrices**: A toeplitz matrix $\mathbf{T} \in \mathbb{C}^{m \times n}$ is a matrix whose entries are constant along each diagonal. A schematic visualization of a toeplitz matrix is shown in Fig 2 **(b)**). They can be parameterized using only their first row and column, allowing them to be encoded in $O(n + m)$ space. Similar to circulant matrices, they support fast $O((n + m) \log(n + m))$ matrix-vector multiplication via FFT.

**Kronecker Product of Matrices**. Kronecker products are another class of structured unrestricted rank matrices that have low storage complexity and admit efficient matrix-vector multiplication. These matrices are obtained using a Kronecker product $\mathbf{A} \otimes \mathbf{B}$ of two matrices $\mathbf{A}$ and $\mathbf{B}$, as shown in Fig 2 **(c)**. We provide more details about these matrices in Appendix A.3.

## 4 LDR-SURMs as General Approximators

In this section, we motivate the usage of structured unrestricted-rank matrices (SURMs) for parameter-efficient fine-tuning. In general, the parameter updates $\Delta\mathbf{W}$ can be arbitrary matrices, and an effective parameterization of $\Delta\mathbf{W}$ should be sufficiently expressive to approximate them. Since we use structured update rank matrices (SURMs) to parameterize $\Delta\mathbf{W}$, we demonstrate that SURMs can approximate various classes of matrices. Without loss of generality, in this section, we assume that all our matrices have real entries and that weight matrices are square ($n = m$).

First, we recall the result from [66], which states that a broad class of low displacement rank matrices, as well as linear combinations of Toeplitz (or their inverses) products[2], can be parameterized as:

$$
\mathbf{W}(\mathbf{G}, \mathbf{H}) = \sum_{i=1}^{r} \mathbf{Z}_1(\mathbf{g}_i)\mathbf{Z}_{-1}(\mathbf{h}_i), \tag{2}
$$

where $\mathbf{G} = [\mathbf{g}_1, ..., \mathbf{g}_r], \mathbf{H} = [\mathbf{h}_1, ..., \mathbf{h}_r] \in \mathbb{R}^{n \times r}$, and $\mathbf{Z}_f(\mathbf{v})$ (for any $f \in \mathbb{R}, v \in \mathbb{R}^n$) is defined as:

$$
\mathbf{Z}_f(\mathbf{v}) = \begin{bmatrix}
v_0 & fv_{n-1} & \cdots & fv_1 \\
v_1 & v_0 & \cdots & fv_2 \\
\vdots & \vdots & \vdots & fv_{n-1} \\
v_{n-1} & \cdots & v_1 & v_0
\end{bmatrix}. \tag{3}
$$

When $f = 1$, $\mathbf{Z}_f(v)$ is a circulant matrix and when $f = -1$, we refer to $\mathbf{Z}_f(v)$ as a skew-circulant matrix. Moreover, $\mathbf{F}$ can be decomposed as follows: $\mathbf{F} = \mathbf{G}\mathbf{H}^\top$ for $\mathbf{G} = [\mathbf{g}_1, ..., \mathbf{g}_r], \mathbf{H} = [\mathbf{h}_1, ..., \mathbf{h}_r] \in \mathbb{R}^{n \times r}$. One can think about rank $r$ of $\mathbf{F}$ of controlling how "structured" $\mathbf{W}$ is.

---

[2]$\mathbf{M}_1 \cdot ... \cdot \mathbf{M}_t$ for $r \geq 2t$ and where each $\mathbf{M}_i$ is a Toeplitz matrix or its inverse.

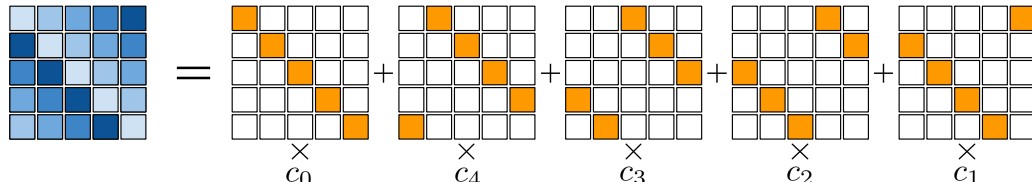

Figure 3: A circulant matrix with the first column given by a vector $(c_0, c_1, c_2, c_3, c_4)$ can be re-written as a linear combination of the orthogonal base circulant matrices (5 matrices with orange-entries corresponding to one and other to zero). Such a closed-form decomposition is in general not possible for matrices $\mathbf{W}(\mathbf{G}, \mathbf{H})$ and thus optimal approximators are found by gradient-descent.

From the above result, we see that $\mathbf{W}(\mathbf{G}, \mathbf{H})$ is the most expressive parameterization among the ones discussed so far. To understand how they fare with SURMs in practice, we evaluate their approximation qualities in two settings: **(a)** comparing $\mathbf{W}(\mathbf{G}, \mathbf{H})$ with circulant and Toeplitz matrices, and **(b)** comparing circulant and Toeplitz matrices with low-rank matrices.

## 4.1 Comparing $\mathbf{W}(\mathbf{G}, \mathbf{H})$ with Circulant and Toeplitz Matrices

We test the approximation capabilities of matrices $\mathbf{W}(\mathbf{G}, \mathbf{H})$ (Eq. 2) and compare it with popular classes of SURMs: circulant and Toeplitz matrices (Section 3). Specifically, we use these structured matrices to approximate three broad classes of matrices: **(a)** random, **(b)** near-low-rank, and **(c)** near-low-intrinsic-rank. We denote the ground-truth matrix that we try to approximate as $\mathbf{M} \in \mathbb{R}^{100 \times 100}$ and parameterized structured matrix as $\mathbf{A}$ (see more details about the setup in Appendix A.7). For all matrices, we obtain the parameters of $\mathbf{A}$ using gradient descent on the loss function: $\|\mathbf{A} - \mathbf{M}\|_\mathrm{F}^2$. [3]

In Figure 5, we report the relative Frobenius norm error during training for different settings. In Fig 5 (top left), we use $\mathbf{W}(\mathbf{G}, \mathbf{H})$ with different $r$ (rank of $\mathbf{F}$ in Eq. 1) is used to approximate random matrices. While the best approximations are achieved for larger values of $r$ (specifically, $r = 20$), it is interesting to note that the final error does not decrease monotonically with increasing $r$. For the remaining class of matrices $\mathbf{M}$, which are close to low-rank and therefore easier to approximate, we experiment with smaller values of $r$ and report the results in Figure 5 (left column, middle and bottom). In this case, we observe that the three top-performing approximators $\mathbf{W}(\mathbf{G}, \mathbf{H})$ were trained with $r = 1, 2, 4$. These results indicate that for more structured ground truth matrices (even if they are not necessarily low-rank), LDRMs with a very low rank for the corresponding $\mathbf{F}$ are sufficient.

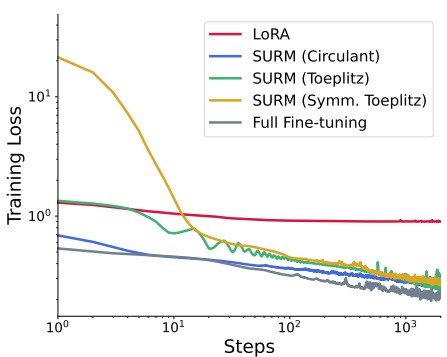

Figure 4: Fitting the pinwheel dataset with a frozen embedding layer using various SURM-based PEFT methods and LoRA.

Motivated by the results showing that LDRMs with low $r$ can serve as effective approximators, we use circulant and Toeplitz matrices to approximate near-to-low-rank and low-intrinsic-rank matrices. In the three plots shown in Fig. 5 (right column), we observe that approximations using Toeplitz matrices (using twice as many parameters as circulant matrices) offer negligible gains and are only beneficial in the near-low-rank case. For the low-intrinsic case, circulant matrices outperform Toeplitz ones. Overall, circulant matrices with few parameters achieve strong performance in this setting.

## 4.2 Comparing Low Rank with Circulant and Toeplitz Matrices

In this section, we focus on the difference in approximation qualities between low rank matrices and the circulant and Toeplitz matrices under a *fixed* parameter budget. We use the following settings:

---

[3]Please note that for circulant and Toeplitz matrices, it is also possible to obtain a closed-form solution for the matrix (see Fig 3 & Appendix A.6).

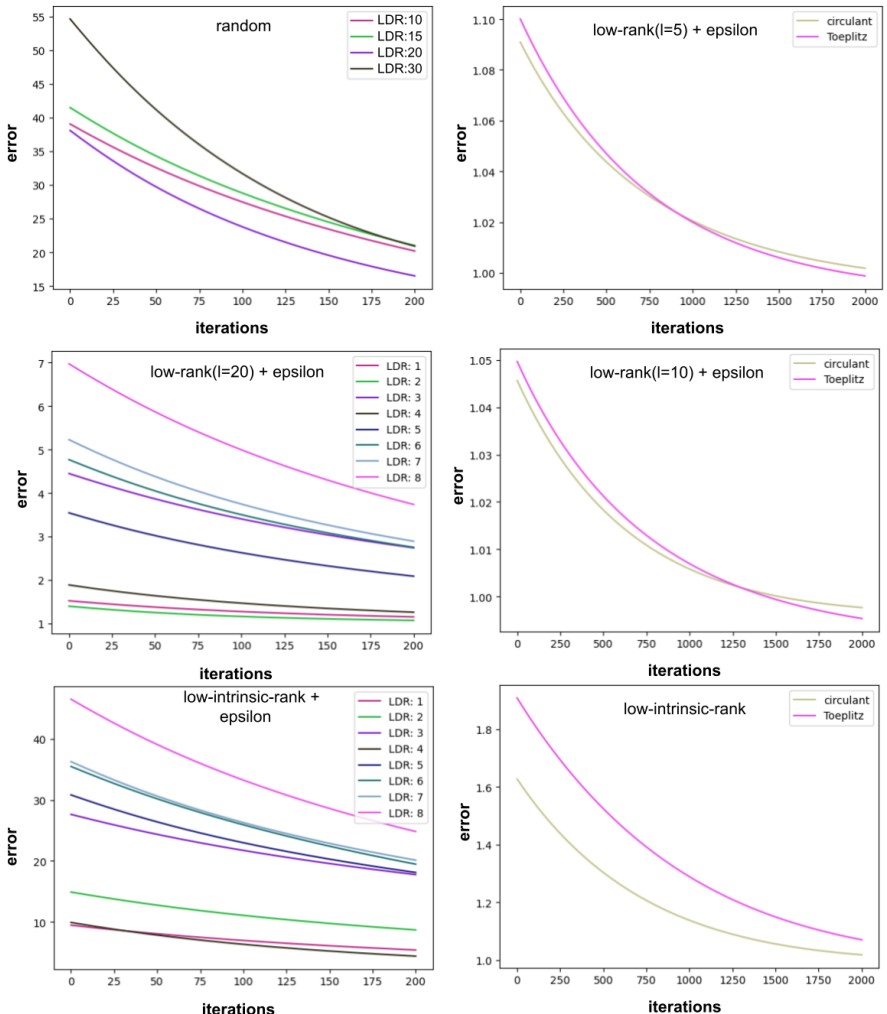

Figure 5: Illustration of the approximation capabilities of different LDRMs. The $y$-axis depicts the relative Frobenius norm error $\|\mathbf{A} - \mathbf{M}\|_{\mathrm{F}}/\|\mathbf{M}\|_{\mathrm{F}}$ between the groundtruth $\mathbf{M}$ and the approximator $\mathbf{A}$. (*Left Column Top*): We approximate a random Gaussian matrix $\mathbf{M}$ with matrices $\mathbf{W}(\mathbf{G}, \mathbf{H})$ using different $r$ (LDR: $r$). (*Left Column Middle*) We approximate near-low-rank matrices $\mathbf{M}$ using smaller values of $r$. (*Left Column Bottom*): Similar setup to approximate near-low-intrinsic-rank matrices $\mathbf{M}$. (*Right Column*): We perform analogous studies with circulant and Toeplitz matrices, where the ground truth has low rank or low-intrinsic rank.

**Approximating Symmetric Positive Definite Matrices**. We use a PSD matrix $\mathbf{M} \in \mathbb{R}^{50 \times 50}$ with $L^2$-normalized rows in this experiment. We compare the errors to approximate $\mathbf{M}$ using circulant, (symmetric) Toeplitz, low-rank matrices, and Kronecker product of two matrices. We use a fixed parameter budget and repeat this experiment 10 times. We report the results in Figure 1 (left). For *each* of these 10 trials, we observe that the circulant and Toeplitz achieve the lowest error, therefore the best approximation quality (see more details in Appendix A.7).

**Fitting a Toy Dataset**. We create a synthetic pinwheel dataset with 5 spokes as shown in Figure 9 (left). We fit this dataset using a simple neural network with one hidden layer with a matrix $\mathbf{W} \in \mathbb{R}^{64 \times 64}$. In this experiment, we replace $\mathbf{W}$ with a rank 1 LoRA, a circulant, a symmetric Toeplitz, and a Toeplitz matrix ( with all parameterizations having the same number of training parameters). In Fig 4), we report the training loss curves of this experiment. We observe that the LoRA layer struggles to fit the data whereas the LDRMs show similar performance to full fine-tuning (Fig 4). These results show the impressive expressive power of these matrices. Therefore, we conclude that LDRMs with particularly low displacement rank serve as good approximators for various matrices. We perform additional experiments and provide more details in Appendix B.

Table 1: ViT-experiments : Baseline numbers are taken from [24]. The best numbers are highlighted in **bold** and the second-best numbers are underlined. Hyperparameter settings are followed from [24]. We find that SURM consistently outperform very strong baselines with **2-3x** reduction in parameters.

| Method | # Param ($\times 10^6$) | ViT-B | | | | | CLIP | | | | |
|---|---|---|---|---|---|---|---|---|---|---|---|
| | | CIF-10 | CIF-100 | SUN397 | DTD | STL10 | CIF-10 | CIF-100 | SUN397 | DTD | STL10 |
| Fine-tuning | 86.6 | 99.0 | 92.4 | 75.0 | 72.4 | 99.6 | 97.7 | 85.4 | 73.8 | 79.0 | 99.7 |
| Attn. Tuning | 28.4 | 93.9 | 85.7 | 73.8 | 69.2 | 99.2 | 96.8 | 81.8 | 73.1 | 75.0 | 97.6 |
| Trans. Probing | 3.2 | 86.9 | 86.9 | 76.7 | 72.0 | 99.0 | 95.6 | 80.1 | 74.3 | 75.9 | 98.5 |
| Linear Probing | **0.049** | 96.3 | 87.7 | 70.1 | 72.7 | 98.7 | 94.8 | 80.1 | 72.4 | 75.4 | 98.4 |
| BitFit | 0.358 | 92.3 | 81.0 | 71.8 | 72.6 | 99.0 | 92.1 | 76.0 | 70.8 | 75.9 | 98.8 |
| Adapter | 1.505 | 98.4 | 90.6 | 74.2 | 71.0 | 99.3 | 94.7 | 81.4 | 77.1 | 78.0 | 99.0 |
| AdapterDrop | 0.174 | 96.8 | 88.4 | 72.3 | 70.2 | 99.6 | 93.3 | 78.3 | 71.4 | 77.1 | 98.0 |
| LoRA | 0.219 | **98.7** | 90.6 | 73.6 | 70.4 | 99.4 | 95.1 | 78.1 | 80.8 | 78.1 | 99.2 |
| LoRA-Fix | 0.148 | 96.2 | 88.3 | 72.0 | 65.5 | 99.0 | 92.5 | 77.1 | 60.0 | 77.7 | 88.6 |
| LN Tuning | 0.075 | 92.2 | 71.7 | 72.0 | 69.0 | 98.8 | 82.5 | 76.6 | 66.7 | 72.4 | 99.1 |
| LePE | 0.167 | 93.7 | 90.8 | 73.2 | 69.8 | 99.1 | 95.1 | 78.9 | 68.0 | 75.4 | 98.0 |
| RPB | 0.145 | 96.7 | 87.0 | 72.4 | 70.4 | 98.9 | 94.7 | 77.1 | 68.4 | 75.2 | 97.9 |
| KAdaptation | 0.114 | 97.9 | **91.2** | 75.1 | 71.4 | 99.4 | 95.9 | 84.8 | 74.0 | 78.1 | 99.2 |
| SURM (*Kronecker*) | 0.055 | 98.3 | 89.9 | 78.6 | 75.4 | 99.6 | **97.1** | **85.0** | 80.7 | **79.0** | 99.2 |
| SURM (*Toeplitz*) | 0.055 | 98.5 | 90.2 | 79.1 | 75.6 | 99.7 | 97.1 | 84.5 | 80.9 | 77.9 | 99.0 |
| SURM (*Circulant*) | 0.055 | 98.0 | 90.7 | **80.5** | **75.7** | **99.8** | 97.0 | 84.6 | **81.1** | 78.6 | **99.3** |

# 5 Integration of SURMs with PEFT

Motivated by the results from the previous section, we use SURMs as drop-in replacements for various PEFT methods. In this section, we present the integration of SURMs in two popular classes of PEFT methods: LoRA and Adapters.

## 5.1 Integration of SURMs in LoRA

LoRA [26] uses a low rank matrix to parameterize the weight matrix updates. Formally, given a pre-trained weight matrix $\mathbf{W}$, then the updated matrix is $\widehat{\mathbf{W}} = \mathbf{W} + \alpha \Delta \mathbf{W}$, where $\Delta \mathbf{W} = \mathbf{A} \mathbf{B}^\top$ and $\mathbf{A} \in \mathbb{R}^{m \times r}, \mathbf{B} \in \mathbb{R}^{n \times r}$ for $r \ll \min(m, n)$ and $\alpha$ is a fixed scaling parameter. For efficient training, $\Delta \mathbf{W}$ needs to be initialized as a zero-matrix. LoRA performs this by choosing initializing $\mathbf{A}$ to be the zero matrix and $\mathbf{B}$ to be a random matrix. In this work, we propose to parameterize $\Delta \mathbf{W}$ using structured unrestricted rank matrices. Next, we provide the details of parameterizing $\Delta \mathbf{W}$ using different SURM matrices (assuming $m = n$ for simplicity).

**Circulant Matrices**. In this setting, we parameterize the updates as: $\mathbf{W} = \mathbf{C}_1 \odot \mathbf{C}_2$, where $\mathbf{C}_i \in \mathbf{R}^{n \times n}$ are circulant matrices encoded using a $n$-dimensional vector, $\mathbf{r}_i \in \mathbb{R}^n$. We use Hadamard products ($\mathbf{C}_1 \odot \mathbf{C}_2$) instead of conventional matrix products as it can be computed efficiently. The construction of Hadamard products which is O($n$) is quicker than the process involved in efficient multiplication (which is O($n\log(n)$). To enable zero-initialization, we initialize $\mathbf{C}_1$ as a zero-vector and $\mathbf{C}_2$ as a random-vector. Additionally, this approach does not compromise the expressiveness of the network, as the result of the Hadamard product is also a circulant matrix.

**Toeplitz Matrices**. Similar to the previous setting, we use two Toeplitz matrices to parameterize: $\Delta \mathbf{W} = g(\mathbf{T_2}, g(\mathbf{T_1}, \mathbf{x}))$, where $\mathbf{T_1}, \mathbf{T_2} \in$ are Toeplitz matrices, and $g$ is the operator that allows efficient matrix-vector multiplication with Toeplitz matrices (see Appendix A.3 ). Each Toeplitz matrix $\mathbf{T} \in \mathbb{R}^{n \times n}$ is parameterized using an $n$-dimensional vector $\mathbf{r}$ encoding its first row and an $n$-dimensional vector $\mathbf{c}$ encoding its first column ($2n - 1$ total parameters). This formulation leads to the $4n - 2$ trainable parameters. To further reduce this number, we constrain the $\mathbf{T_1}, \mathbf{T_2}$ to be symmetric, reducing the total number of trainable parameters to $2n$. To enable zero-initialization, we initialize $\mathbf{T}_1$ as a zero matrix and $\mathbf{T}_2$ is randomly initialized.

**Kronecker Product of Matrices**. In this setting, we parameterize: $\Delta \mathbf{W} = \mathbf{A} \otimes \mathbf{B}$, where $\mathbf{A} \in \mathbb{R}^{r_1 \times r_2}, \mathbf{B} \in \mathbb{R}^{\frac{n}{r_1} \times \frac{n}{r_2}}$. The hyperparameters $r_1, r_2$ allow us to control the trainable parameter count and the rank of $\Delta \mathbf{W}$. In contrast to low-rank matrix updates, we can create matrices $\Delta \mathbf{W}$ of fairly large ranks while keeping the number of trainable parameters small (see Appendix A.2). To enable zero-initialization, we set $\mathbf{A}$ as a zero matrix and $\mathbf{B}$ as a random matrix.

Table 2: Results on the VTAB-1k benchmark. Baseline numbers are taken from [30] and [50]. Best numbers are highlighted in **bold** and the second-best numbers are underlined. We observe that SURM is one of the top-performing PEFT methods on almost all datasets.

| Method | # Params $(\times 10^6)$ | CIF-100 | Cal-101 | DTD | F-102 | Pets | SVHN | Sun397 | Cam. | EuroSAT | Res-45 | Retino. |
|---|---|---|---|---|---|---|---|---|---|---|---|---|
| Fine-tuning | 86.6 | 68.9 | 87.7 | 64.3 | 97.2 | 86.9 | 87.4 | 38.8 | 79.7 | 95.7 | 84.2 | 73.9 |
| Linear | **0.049** | 64.4 | 85.0 | 63.2 | 97.0 | 86.3 | 36.6 | 51.0 | 78.5 | 87.5 | 68.5 | 74.0 |
| BitFit | 0.013 | 72.8 | 87.0 | 59.2 | 97.5 | 85.3 | 59.9 | 51.4 | 78.7 | 91.6 | 72.9 | 69.8 |
| VPT-Shallow | 0.063 | 77.7 | 86.9 | 62.6 | 97.5 | 87.3 | 74.5 | 51.2 | 78.2 | 92.0 | 75.6 | 72.9 |
| VPT-Deep | 0.531 | 78.8 | 90.8 | 65.8 | 98.0 | 88.3 | 78.1 | 49.6 | 81.8 | 96.1 | 83.4 | 68.4 |
| Adapter | 0.157 | 69.2 | 90.1 | 68.0 | 98.8 | 89.9 | 82.8 | 54.3 | 84.0 | 94.9 | 81.9 | 75.5 |
| AdaptFormer | 0.157 | 70.8 | 91.2 | 70.5 | 99.1 | 90.9 | 86.6 | 54.8 | 83.0 | 95.8 | 84.4 | **76.3** |
| LoRA | 0.295 | 67.1 | 91.4 | 69.4 | 98.8 | 90.4 | 85.3 | 54.0 | 84.9 | 95.3 | 84.4 | 73.6 |
| NOAH | 0.361 | 69.6 | **92.7** | 70.2 | 99.1 | 90.4 | 86.1 | 53.7 | 84.4 | 95.4 | 83.9 | 75.8 |
| Fact-TK$_{\leq 32}$ | 0.069 | 70.6 | 90.6 | 70.8 | 99.1 | 90.7 | 88.6 | 54.1 | 84.8 | **96.2** | 84.5 | 75.7 |
| SSF | 0.240 | 69.0 | 92.6 | **75.1** | 99.4 | 91.8 | 90.2 | 52.9 | **87.4** | 95.9 | **87.4** | 75.5 |
| RepAdapter | 0.110 | 70.7 | 91.6 | 72.5 | 99.1 | 91.3 | 88.5 | 54.2 | 84.1 | 95.7 | 85.1 | 74.6 |
| SURM (*Kronecker*) | 0.055 | 79.6 | 88.7 | 73.1 | 99.1 | 92.5 | 74.8 | 54.7 | 82.2 | 94.3 | 81.9 | 75.4 |
| SURM (*Toeplitz*) | 0.055 | 79.5 | 88.9 | 72.7 | 99.1 | 91.5 | 74.7 | 55.8 | 83.6 | **96.2** | 82.2 | 76.0 |
| SURM (*Circulant*) | 0.055 | **80.6** | 87.5 | 74.7 | **99.5** | **93.3** | 74.9 | **57.1** | 85.3 | 96.0 | 83.7 | 75.4 |

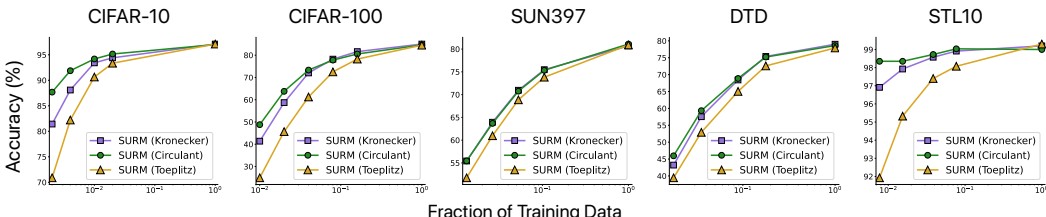

Figure 6: **Low Resource Training**. Accuracy of SURM CLIP-ViT models as a function of the training data fraction. The results show that SURM can achieve comparable accuracy with as low as $\sim 2\%$ of the training data for easier tasks like CIFAR10 and $\sim 20\%$ for harder tasks like SUN397.

In all the above settings, it is possible to increase the number of training parameters by relaxing the structure of the matrix, $\Delta \mathbf{W}$. This can be performed by introducing more matrices in the product chains, utilizing asymmetric Toeplitz matrices, adjusting the sizes of factors in the Kronecker product, or employing sums of such matrices. Another way to enhance layer expressiveness is by experimenting with combinations of different LDRMs, such as mixing circulant and skew-circulant matrices. A broad class of matrices, including low-rank ones, can be represented as sums of these matrices (see Theorem A.2 and the subsequent discussion).

## 5.2 Integration of SURMs in Adapters

Adapters [25] are small bottleneck networks into Transformer layers as shown below:

$$\mathbf{Y} = \mathbf{X} + \sigma(\mathbf{X}\mathbf{B})\mathbf{A}, \tag{4}$$

where $\sigma(\cdot)$ is a non-linear activation function, $\mathbf{X} \in \mathbb{R}^{b \times s \times n}$ represents input to the layer ($b$: batch size, $s$: sequence length), $\mathbf{A} \in \mathbb{R}^{r \times n}, \mathbf{B} \in \mathbb{R}^{n \times r}$ are low-rank matrices ($r \ll n$) and $\mathbf{Y}$ is the output of the layer. For simplicity, layer norms and bias terms are not included in the equation. We use SURMs as a drop-in replacement for matrices $\mathbf{A}$ and $\mathbf{B}$. Next, we will provide details of integrating SURMs within the adapter setting. Additional details are provided in Appendix A.5.

**Circulant Matrices**. In this setting, we apply two circulant matrices $\mathbf{C}_1, \mathbf{C}_2$ (encoded by $\mathbf{r}_1, \mathbf{r}_2$), resulting in the adapter block: $\mathbf{Y} = \mathbf{X} + \sigma(f(\mathbf{r_1} \odot \mathbf{r_2}, \mathbf{X})) + \mathbf{b}$, where $f$ is an operator that efficiently computes the matrix multiplication between input $\mathbf{X}$ and the circulant matrix encoded by the vector $\mathbf{r}_1 \circ \mathbf{r}_2$ (Appendix A.3). To enable zero-initialization, the vector $\mathbf{r}_1$ is initialized randomly while $\mathbf{r}_2$ and $\mathbf{b}$ are initialized as zero vectors.

**Toeplitz Matrices**. In this setting, we use two symmetric Toeplitz matrices $\mathbf{T_1}, \mathbf{T_2}$, where $\mathbf{T_1}$ and $\mathbf{b}$ is initialized as a zero vector and $\mathbf{T_2}$ is initialized randomly. We then define the adapter layer as: $\mathbf{Y} = \mathbf{X} + \sigma(g(\mathbf{T_1}, g(\mathbf{T_2}, \mathbf{X}))) + \mathbf{b}$, where $g$ is an operator that efficiently computes the matrix multiplication between an input $\mathbf{X}$ and a Toeplitz matrix.

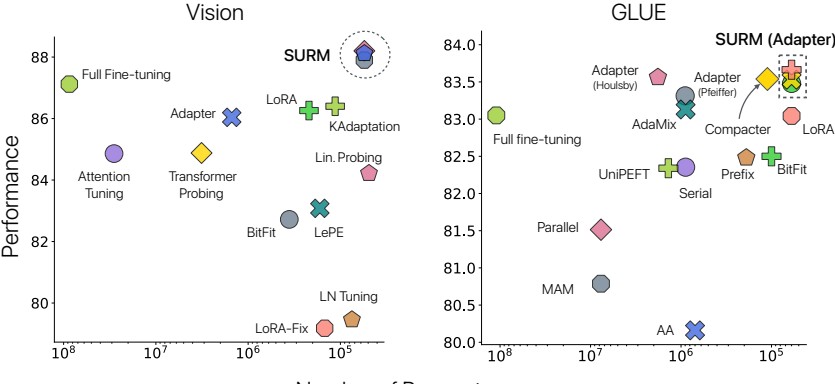

Figure 7: **Left:** Tradeoff between performance and parameter count for various PEFT methods. We report the average results across 5 image datasets using ViT-B (complete results in Table 1). **Right:** Average performance across GLUE benchmark (see complete results in Table 5). SURMs appear in the top right corner and perform best among various strong baseline methods in both settings.

**Kronecker Product of Matrices.** In this case, we rewrite Equation 4 as: $\mathbf{Y} = \mathbf{X} + \sigma(\mathbf{X}(\mathbf{B} \otimes \mathbf{A})) + \mathbf{b}$, where $\mathbf{B} \otimes \mathbf{A}$ is the Kronecker product. In this case, $\mathbf{B}$ is initialized randomly and $\mathbf{A}$ and $\mathbf{b}$ are initialized by zeros. In all experiments using SURM-adapters, $\sigma(\cdot)$ is the GeLU non-linearity.

# 6 Experiments

In this section, we show the effectiveness of our proposed methods in a wide range of vision and NLP tasks through extensive empirical studies.

**Image Classification Experiments.** We evaluate SURM on several vision datasets: CIFAR10, CIFAR100 [39], SUN397 [79], DTD [16] and STL10 [17]. We experiment using ViT-B/16 [36] & Clip-ViT-B/16 [63] as base models and inject trainable parameters $\mathbf{Q}, \mathbf{K}, \mathbf{V}$ matrices in the LoRA setting. We report the results using ViT$_{base}$ are presented in Table 1 (left). We observe that SURM consistently outperforms **12** baseline methods (that use up to **10x** parameters). On three out of the five tasks, SURMs emerge as the top performers, surpassing LoRA by a margin of up to **5-7**% while achieving competitive performance on the remaining tasks. We report the results using Clip-ViT in Table 1 (right). In this setting, SURM is among the top two methods across all 5 tasks. SURM also uses fewer trainable parameters, reducing them by **3.65x** compared to LoRA and **2.4x** compared to LoRA-Fix.

*Low Data Regime.* We evaluate SURM in low data regime using VTAB-1k datasets [85] and the ViT model. VTAB-1k is a diverse collection of vision datasets with only 1000 training examples. We focus on the NATURAL and SPECIALIZED subsets of VTAB. In Table 2, we observe that SURMs are among the top 2 methods on **10** datasets while being competitive on the remaining tasks.

We evaluate different variants of SURM and train it on a varying fraction of data on 5 datasets using Clip-ViT model. We report the results in Figure 6. We observe that the circulant SURM works best in low data regime. Furthermore, SURM achieves the performance of full fine-tuning trained on the entire dataset with only a small fraction of the data. For more challenging datasets like SUN397, we achieve comparable accuracy using approximately 20% of the training data, while for datasets such as CIFAR10 and STL10, only about 2% is needed.

*Large Data Regime.* We perform experiments to show that SURM generalizes well in large data regimes. On ImageNet [18] and iNat2021 [72], SURM achieves performance comparable to full fine-tuning while using only **0.06**% of the training parameters (see detailed results in Appendix B.2).

**NLP Experiments.** We extensively evaluate SURM models on the GLUE benchmark [74] using BERT$_{base}$ [20]. We compare with different adapter baselines and 11 other PEFT techniques. These include full-finetuning, *Adapter (Houlsby)* and *Adapter (Pfeiffer)*, among others (we report the corresponding results from [59]). BiTFit results are taken from [84] (except QQP numbers which are obtained from [9]) and the numbers for *AA*-adapters from [53]. Prefix, Serial, AdaMix, UniPELT,

Table 3: Image Segmentation results on the Synapse multi-organ segmentation dataset. SURMs achieve comparable performance with specialized architectures developed for medical imaging while being more parameter efficient.

| Methods | DSC | Aorta | Gallblad. | Kid. (L) | Kid. (R) | Liver | Pancreas | Spleen | Stomach |
|---------|-----|-------|-----------|----------|----------|-------|----------|--------|---------|
| U-Net | 76.85 | 89.07 | **69.72** | 77.77 | 68.60 | 93.43 | 53.98 | 86.67 | 75.58 |
| Att-UNet | 77.77 | **89.55** | 68.88 | 77.98 | 71.11 | 93.57 | 58.04 | 87.30 | 75.75 |
| TransUnet | 77.48 | 87.23 | 63.13 | 81.87 | 77.02 | 94.08 | 55.86 | 85.08 | 75.62 |
| SwinUnet | 79.13 | 85.47 | 66.53 | **83.28** | 79.61 | 94.29 | 56.58 | **90.66** | 76.60 |
| SAMed | **81.88** | 87.77 | 69.11 | 80.45 | 79.95 | **94.80** | **72.17** | 88.72 | **82.06** |
| LORA (rank=1) | 78.26 | 81.86 | 64.54 | 81.97 | **81.18** | 93.79 | 60.80 | 88.33 | 73.64 |
| SURM (*Circulant*) | 80.11 | 83.04 | 64.92 | 81.37 | 80.96 | 94.21 | 69.11 | 88.15 | 79.06 |

Parallel, MAM, and AutoPEFT numbers are taken from [90]. The results for the remaining baselines are replicated by us. More experimental details can be found in Appendix B.

For brevity, we summarize the average performance across 8 tasks for SURM-adapters and compare it to 11 baselines, in Fig 7 (right). We observe that SURM achieve much better performance while using a fraction of the parameters (the complete results are reported in Appendix Table 5). We also observe that SURM (integrated into LoRA) outperforms the baseline LoRA, under the same parameter budget. This shows the effectiveness of using structured matrices as a drop-in replacement for low rank matrices used in LoRA. We further analyze the representations learnt by SURMs and LoRAs (Appendix B.4). We find that LoRA learns weights that are very similar to the pre-trained weights whereas SURM is able to explore a larger parameter space (an observation similar to [91]).

**Large-scale Experiments**. In this setting, we integrate SURMs in LLMs. Specifically, we use matrices of the form $\mathbf{W}(\mathbf{G}, \mathbf{H})$ (as described in Eqn 2) to increase the number of training parameters. We use the experimental setup introduced in [29], where the LLM tries to fit a dataset of UUID pairs using the Llama-2-7B model [69]. This was shown to be a challenging task (UUID prediction is significantly different from the pre-training tasks) that requires higher rank values in LoRA. We report the results in Fig. 8, demonstrating that SURMs is able to fit the data, whereas other methods struggle to do so (see more details in Appendix B.1).

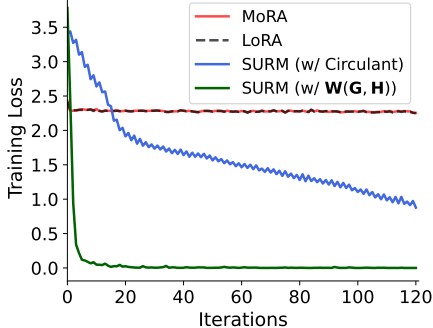

Figure 8: Fitting the UUID dataset using Llama-2-7b. We fit the data using various SURM-based PEFT methods and LoRA.

**Image Segmentation**. Next, we focus on the extremely challenging task of medical image segmentation using Synapse multi-organ segmentation dataset [82]. Segment-Anything-Model (SAM) [34] is used as the foundation model for this task. We follow [86] and adapt the $\mathbf{Q}, \mathbf{V}$ in ViT$_\text{base}$ image encoder in SAM. Finally, in this low data regime, we use Circulant variant of SURM as it is the best performing variant (Fig. 6). We report the Dice similarity coefficient (DSC) metric for each of the 8 organ segmentations and their average (higher is better). For a fair comparison, we include LoRA with rank 1, matching the *exact* parameter count of Circulant. The results are presented in Table 3. We report the baseline performance from [86]. SURMs compare favorably with specialized architectures developed for medical imaging like U-Net, Attention U-Net, Transformer-based U-Net, and the Swin U-Net even though they have significantly higher number of training parameters than our method (see details in Appendix B.1).

## 7 Conclusion

We introduce structured unrestricted-rank matrices (SURMs) as an alternative to low-rank matrices for the parameter-efficient fine-tuning of large Transformer models. In this setting, structured matrices form the cornerstone of a comprehensive framework, offering a solid base for various parameter efficient fine-tuning methods, such as adapters and LoRA, with enhanced efficiency. SURMs improve the overall effectiveness of PEFT, contributing to its efficient integration into diverse models and domains. Based on extensive numerical experiments and theoretical insights, we conclude that the Circulant variant is our most performing variant (in terms of speed and accuracy).

## 8 Author Contributions

AS designed the integration of SURM in Adapters and LoRA and ran the GLUE experiments. AD helped in developing the integration and ran all image experiments. KC came up with the idea of using LDRMs in the context of PEFT. SBRC helped in running various large-scale experiments and writing the manuscript. All authors contributed to the writing of this manuscript.

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

# A   Implementation Details

In this section, we discuss the details of various algorithms and workflows within SURM.

**Contents**

## A.1   Skew-Circulant Matrices

In this section, we introduce another type of structured matrix that is characterized by a linear number of parameters.

**Definition A.1** (Skew-Circulant). A matrix $\mathbf{S} = (s_{jk})_{j,k=0}^{n-1}$ is said to be skew-circulant if $s_{jk} = s_{j-k}$ and $s_{-l} = -s_{n-l}$ for $1 \leq l \leq n-1$.

This matrix can be represented visually as shown below:

$$\mathbf{S} = \begin{bmatrix} v_0 & -v_{n-1} & \cdots & -v_1 \\ v_1 & v_0 & \cdots & -v_2 \\ \vdots & \vdots & \vdots & -v_{n-1} \\ v_{n-1} & \cdots & v_1 & v_0 \end{bmatrix} \tag{5}$$

This matrix is parameterized by a linear number of parameters and also enjoy sub-quadratic time complexity matrix-vector multiplications (see Appendix A.3).

## A.2   Finding the Smallest Number of Training Parameters for Kronecker Layers

Let $\mathbf{W}$ be a $d \times d$ matrix that can be written as $\mathbf{W} = \mathbf{A} \otimes \mathbf{B}$, where $\mathbf{A} \in \mathbb{R}^{m_1 \times n_1}, \mathbf{B} \in \mathbb{R}^{m_2 \times n_2}$. We want to minimize the following objective:

$$m_1 n_1 + m_2 n_2, \text{ subject to } m_1 m_2 = n_1 n_2 = d. \tag{6}$$

We can rewrite the above as : $m_1 = d/m_2$ and $n_1 = d/n_2$. Plugging these back in Eq. 6, we get:

$$\begin{aligned} m_1 n_1 + m_2 n_2 &= \frac{d^2}{m_2 n_2} + m_2 n_2 \\ &= \frac{d^2}{m_2 n_2} + m_2 n_2 - 2d + 2d \\ &= \left( \sqrt{m_2 n_2} - \frac{d}{\sqrt{m_2 n_2}} \right)^2 + 2d \\ &\geq 2d. \end{aligned} \tag{7}$$

The equality is obtained when $\sqrt{m_2 n_2} = d/\sqrt{m_2 n_2}$, thereby satisfying the constraint $m_2 n_2 = m_1 n_1 = d$. Essentially this result shows that we can minimize the number of training parameters when the matrices $\mathbf{A}$ and $\mathbf{B}$ are similarly sized. Furthermore, since both $\mathbf{A}$ (and $\mathbf{B}$) have $d$ training parameters, we can maximize the rank of the matrix if we can make it close to a square matrix (i.e. we choose 2 factors $a, b$ of $d$, such $ab = d$ and $a$ is as close to $b$ as possible). Note that $\text{rank}(\mathbf{A} \otimes \mathbf{B}) = \text{rank}(\mathbf{A})\text{rank}(\mathbf{B})$. Thus, for our experiments with BERT and ViT models, we take $\mathbf{A}$ to be a matrix of size $32 \times 24$ and $\mathbf{B}$ to be of size $24 \times 32$. This choice of matrix shapes allows us to substantially reduce the computational complexity of matrix-vector multiplication (see Section A.3).

## A.3 Efficient Matrix Vector Multiplication by Structured Matrices

One of our main advantages of using structured matrices is that they allow for sub-quadratic vector-matrix multiplications. Matrix vector multiplication by a circulant matrix can be efficiently done via FFT in $O(n \log n)$ time. This is done by the following steps : (a) take the FFT of the input vector $\mathbf{v}$ and the vector representation of the circulant matrix $\mathbf{c}$, and call them $\mathbf{V}$ and $\mathbf{C}$ respectively. (b) Take the inverse Fourier transform of the Hadamard (element-wise) product of $\mathbf{V}$ and $\mathbf{C}$.

For the sake of convenience, let us define this efficient multiplication operator to be $f$. The key insight behind this approach is that the circular convolution in the time domain corresponds to element-wise multiplication in the frequency domain after FFT. By leveraging FFT, the time complexity of the multiplication is reduced from $O(n^2)$ to $O(n \log n)$.

The same ideas extend to the case of Toeplitz matrices, where one can embed the Toeplitz matrix into a circulant matrix and use FFT as before for efficient matrix-vector multiplication. For ease of reference, let us call the function $g$ that embeds the Toeplitz matrix into a circulant matrix and use the function $f$ as described above to compute the matrix-vector product.

Next, we describe how vector multiplication by skew-circulant matrices can be efficiently performed in $O(n \log n)$ time. If the skew-circulant matrix $\mathbf{S}$ is parameterized by the vector $\mathbf{v}$, then the multiplication is given by

$$\mathbf{y} = \mathbf{S}(\mathbf{v})\mathbf{x} = \bar{\boldsymbol{\eta}} \circ \mathrm{ifft}(\mathrm{fft}(\boldsymbol{\eta} \circ \mathbf{v}) \circ \mathrm{fft}(\boldsymbol{\eta} \circ \mathbf{x})) \tag{8}$$

where $\boldsymbol{\eta} = [1, \eta, \eta^2, \cdots \eta^{n-1}]$, and $\eta = (-1)^{\frac{1}{n}} = \exp(i\pi/n)$, the root of negative unity.

For the case of a matrix $\mathbf{W} = \mathbf{AB}$, where $\mathbf{W} \in \mathbb{R}^{m \times n}, \mathbf{A} \in \mathbb{R}^{m \times r}$ and $\mathbf{B} \in \mathbb{R}^{r \times n}$, then multiplication by $\mathbf{v}$ takes $O(r(m + n))$ and one gets computation gains when $r \ll \min\{m, n\}$. Finally, for a Hadamard product of matrices $\mathbf{A} \in \mathbb{R}^{r_1 \times r_2}, \mathbf{B} \in \mathbb{R}^{k_1 \times k_2}, \mathbf{v} \in \mathbb{R}^{r_2 k_2}, (\mathbf{A} \otimes \mathbf{B})\mathbf{v} = \mathrm{vec}(\mathbf{B}r(\mathbf{v})^\top \mathbf{A}^\top)$, where $\mathrm{vec}(\cdot)$ is the vectorization operator that takes a matrix $\mathbf{M} \in \mathbb{R}^{m \times n}$ and converts it to $\mathbb{R}^{mn \times 1}$ column vector by stacking the columns of $\mathbf{M}$ on top of each other and $r$ is the PyTorch style reshape operator that reshapes the vector $\mathbf{v}$ to a matrix of shape $r_2 \times k_2$. Choosing $\max\{r_i, k_i\} \ll r_i k_i$, for $i = 1, 2$, one can substantially reduce the computational complexity.

## A.4 Increasing Number of Training Parameters

In this section, we explain an elegant way to increase the number of training parameters. We use the sum of product of circulant and skew-circulant matrices of the form

$$\mathbf{M} = \sum_{i=1}^{r} \mathbf{A}_i \mathbf{B}_i \tag{9}$$

where $\mathbf{A}_i$ and $\mathbf{B}_i$ is a circulant and a skew-circulant matrix respectively. Each factor of $\mathbf{M}$ has $2n$ parameters thus $\mathbf{M}$ has $2nr$ parameters.

The class of $n \times n$ matrices $\mathbf{M}$ which can be written via Equation 9 is rich and contains many important classes of matrices.

**Theorem A.2** (Expressivity). *The set of matrices $\mathbf{M}$ which can be written as in Equation 9 contains:*

- *All $n \times n$ Circulant and Skew-Circulant matrices for $r \geq 1$*

- *All $n \times n$ Toeplitz and Inverses of Toeplitz matrices for $r \geq 2$.*

- *All $n \times n$ matrices for $r = n$.*

- *All linear combinations of the form $\sum_{j=1}^{p} \beta_i \mathbf{A}_1^{(j)} \cdots \mathbf{A}_t^{(j)}$ where $r \geq 2tp$, and $\mathbf{A}$ is either a Toeplitz or the inverse of a Toeplitz matrix.*

This is Theorem 3.1 in [66]. Efficient multiplication by matrices of this form can be done in sub-quadratic time by simply combining the results from Sec A.3.

Moreover, we note that by choosing a slightly different parameterization of displacement operators, one can obtain low rank matrices and orthogonal polynomial transforms, including the Discrete Fourier and Cosine Transforms (see Proposition 2 in [68]).

Thus our framework encompasses many important classes of matrices including *low rank* matrices and thus generalizes LoRA.

### A.5 Integration of SURMs in Adapters

In this section, we provide additional details about SURMintegration in Adapters.

For simplicity, we follow the Houlsby configuration [25], but our work is also readily applicable in Pfeiffer configuration as well [59]. Recall the definition of Adapter layers:

$$\mathbf{Y} = \mathbf{X} + \sigma(\mathbf{X}\mathbf{B})\mathbf{A}, \tag{10}$$

where $\sigma(\cdot)$ is a non-linear activation function applied point-wise, $\mathbf{X} \in \mathbb{R}^{b \times s \times n}$ represents input to the layer ($b$ is the batch size and $s$ is the sequence length), $\mathbf{A} \in \mathbb{R}^{r \times n}, \mathbf{B} \in \mathbb{R}^{n \times r}$ are two low-rank matrices ($r \ll n$) and $\mathbf{Y}$ is the output of the layer. Similar to LoRA, matrix $\mathbf{B}$ is initialized randomly, whereas $\mathbf{A}$ is initialized as a zero-matrix. For convenience, layer norms and bias terms are not included in the equation.

SURMs can be used in place of low rank $\mathbf{A}$ and $\mathbf{B}$. The integration and design choices of various LDRs in this setting mimic that of LoRA.

**Circulant Matrices**. Similar to the LoRA setting, we apply two circulant matrices $\mathbf{C}_1, \mathbf{C}_2$, resulting in the following equation of the adapter block:

$$\mathbf{Y} = \mathbf{X} + \sigma(f(\mathbf{r_1} \odot \mathbf{r_2}, \mathbf{X})) + \mathbf{b}, \tag{11}$$

where $f$ is an operator multiplying input matrix $\mathbf{X}$ with the circulant matrix obtained by multiplying two circulant matrices encoded by $\mathbf{r}_1$ and $\mathbf{r}_2$ (Appendix A.3). The vector $\mathbf{r}_1$ is initialized randomly while $\mathbf{r}_2$ and $\mathbf{b}$ are initialized as zero vectors. Note that we apply the non-linearity after we multiply $\mathbf{X}$ with both the circulant matrices. This may hurt the expressiveness of the network but improves computational complexity. Moreover, we only need to save one vector defining the first row of a *circulant* matrix and not both: $\mathbf{r}_1$ and $\mathbf{r}_2$. This results in lower storage costs and faster deployment. This design choice works well in practice as evidenced from the results on the GLUE benchmark (see Table 5).

**Toeplitz Matrices**. Similar to the case of Toeplitz matrices within LoRA, we use two symmetric Toeplitz matrices $\mathbf{T_1}, \mathbf{T_2}$, where $\mathbf{T_1}$ and $\mathbf{b}$ is initialized as a zero vector and $\mathbf{T_2}$ is initialized randomly. We then define the adapter layer to be:

$$\mathbf{Y} = \mathbf{X} + \sigma(g(\mathbf{T_1}, g(\mathbf{T_2}, \mathbf{X}))) + \mathbf{b}. \tag{12}$$

The position of the non-linear mapping $\sigma$ is chosen such that we can merge the two trained matrices resulting in smaller storage costs and fast deployment.

Finally, note that the Toeplitz variant is slower than the circulant variant, as it requires two applications of the fast matrix-vector operator, whereas the circulant variant requires only one.

### A.6 Computing Approximations Using LDR Matrices

In this section, we show how we can approximate any matrix $\mathbf{D} \in \mathbb{R}^{n \times n}$ using Circulant, Toeplitz matrices, and symmetric Toeplitz matrices. We note that each class of structured matrices forms a vector space. Therefore, finding the closest point in the appropriate subspace becomes a convex optimization problem and is given by the orthogonal projection onto the basis vectors of the subspace. More explicitly, if $\{\mathbf{e_1}, \cdots, \mathbf{e_n}\}$ are a set of orthogonal vectors spanning a subspace $\mathbf{W}$, then the closest vector to $\mathbf{v}$ in $\mathbf{W}$ is given by

$$\hat{\mathbf{v}} = \frac{(\mathbf{v}, \mathbf{e_1})}{\|\mathbf{e_1}\|^2}\mathbf{e_1} + \ldots + \frac{(\mathbf{v}, \mathbf{e_n})}{\|\mathbf{e_n}\|^2}\mathbf{e_n}. \tag{13}$$

The space of circulant matrices has dim $n$, so spanned by the orthogonal set $\{(1, \cdots, \cdots 0), (0, \cdots 1, \cdots 0), (0, \cdots, \cdots 1)\}$. Using the above formula, one can write down a simplified expression of the circulant matrix as $\hat{\mathbf{C}} := (\hat{c}_1, \cdots \hat{c}_n)$ that approximates $\mathbf{D}$

$$\hat{c}_1 = \frac{1}{n}\sum_{j=1}^{n} d_{jj}, \quad \hat{c}_k = \frac{1}{n}\left\{\sum_{j=1}^{k-1} d_{j(1+j+n-k)} + \sum_{j=k}^{n} d_{j(j-k+1)}\right\}, \text{ where } k = \{2, \ldots, n\}.$$

Note that the same set as before spans the space of symmetric Toeplitz matrices. This yields a compact formula for the approximating Toeplitz matrix:

$$\hat{\mathbf{T}} := \left(\frac{1}{n}\sum_{i=1}^{n} a_{i,i}\right)\mathbf{I}_n + \left(\frac{1}{n-1}\sum_{i=1}^{n-1} a_{i,i+1}\right)\mathbf{M}_2 + \left(\frac{1}{n-2}\sum_{i=1}^{n-2} a_{i,i+2}\right)\mathbf{M}_3 + \cdots + a_{1,n}\mathbf{M}_n,$$

where $\mathbf{M}_i$ is the symmetric Toeplitz matrix generated by the $i$-th element in the set above. Finally the set $\{((1,0,\cdots,0),(0,\cdots,0)),\cdots((0,\cdots,1,\cdots,0),(0,\cdots,0)),((0,\cdots,0),(0,\cdots,1,\cdots,0)\,\}$ spans all Toeplitz matrices where the first element in each tuple denotes the first row and the second element the first column. Note that since the $a_11$ entry is shared by both first row and column we treat the first vector as $n$-dimensional vector and the second as $n-1$ dimensional vector. Thus the dimension of the space is $2n-1$. Using FFT and the projection formula, one can compute the approximation by a Toeplitz matrix.

## A.7  Additional Details on Approximation Errors by LDR

In this section, we present additional details on the various experiments on approximation by LDR matrices presented in Section 4.1.

- **Random:** The first class, with entries taken independently at random from $\mathcal{N}(0,1)$, represents a completely unstructured family.

- **Near-low rank:** Each matrix from the second class was chosen from the distribution: $\mathbf{G}\mathbf{H}^\top + \epsilon\mathbf{R}$, where $\mathbf{G},\mathbf{H} \in \mathbb{R}^{n\times r}$ for $r \ll n$, $\mathbf{R} \in \mathbb{R}^{n\times n}$, $\epsilon = 0.05$, and the entries of $\mathbf{G},\mathbf{H},\mathbf{R}$ are taken independently at random from $\mathcal{N}(0,1)$.

- **Near-low intrinsic rank:** Matrices from the third class are constructed as follows. First we sample: $t_0,...,t_{n-1} \overset{\text{iid}}{\sim} \mathcal{N}(0,1)$. The $i$-th row of the resulting matrix is of the form: $(\sin(1 \cdot t_i), \sin(2 \cdot t_i),...,\sin(n \cdot t_i)) + \mathbf{g}_i$, wheare either all $\mathbf{g}_i$ are zero-vectors or they are taken independently at random from $\epsilon * \mathcal{N}(0,\mathbf{I}_n)$. Note that even though that matrix is not necessarily low-rank, it is taken from the vicinity of the $n$-dimensional manifold, since it is fully determined by the sampled tuple $(t_0,...,t_{n-1})$. Matrices from all the classes are taken from $\mathbb{R}^{100\times100}$.

**Optimizing Circulant and Toeplitz Matrices**. In general, an optimal approximation (e.g. with respect to the Frobenius norm as a distance) of a given matrix by a matrix $\mathbf{W}(\mathbf{G},\mathbf{H})$ is not given by the closed-form expression. Thus we will thus construct good-quality approximators via gradient-based optimization (see: Sec. 4.1).

**Details on Approximation Experiments in Section 4.2**. Now we provide additional details on the experiments that explicitly compare LDRMs with low-rank matrices. For these experiments, we construct a PSD matrices $\mathbf{M} \in \mathbb{R}^{50\times50}$ with $L^2$ normalized rows. We fix a parameter budget of $n = 50$. The low-rank approximation, in that case, becomes an outer product by a vector $\mathbf{v}$. For the Kronecker product, we choose a factor $\mathbf{A} \in \mathbb{R}^{10\times5}$. To maintain the parameter budget, the other factor becomes $\mathbf{A}^\top$. If $\hat{\mathbf{M}}$ is the approximating matrix, then we define error $= ||\hat{\mathbf{M}} - \mathbf{M}||_F$, where $||\cdot||_F$ is the Frobenius norm.

We use the closed-form formula for the optimal circulant and symmetric Toeplitz matrices approximating $\mathbf{M}$ and use gradient descent to find the optimal low-rank matrix and Kronecker product of matrices. We use a learning rate of $0.1$ while computing the optimal low-rank matrix and the Kronecker product of matrices.

## A.8  Invertible Toeplitz Matrices

Inverses of Toeplitz matrices can be effectively found [40]. We recall the celebrated result of Gohberg and Semencul.

**Theorem A.3.** *Let* $\mathbf{A} := (a_{p-q})_{p,q=1}^n$ *be a Toeplitz matrix. If the following systems of equations*

$$\sum_{q=1}^{n} a_{p-q}x_q = \delta_{p,1}, \sum_{q=1}^{n} a_{p-q}y_q = \delta_{p,n}, \text{ where } p = \{1,2\ldots n\}$$

*is solvable and* $x_1 \neq 0$*, then* $\mathbf{A}$ *is invertible.*

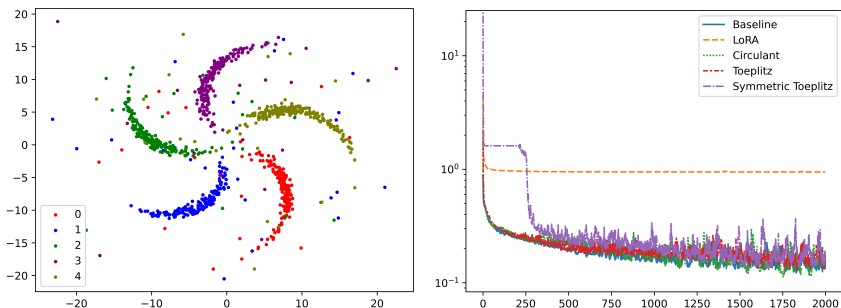

Figure 9: Experiment on fitting the pinwheel dataset. **Left:** Visualization of the pinwheel dataset. **Right:** Results of fitting the pinwheel dataset using regular training, where all network parameters are trained. A network with a low-rank hidden layer matrix struggles to fit the data, while those with SURMmatrices achieve a successful fit.

In our case, we consider only symmetric Toeplitz matrices. Thus the above equation really boils down to solving the first system of equations as the next system can be solved by using the first, i.e. by setting $x_{n-i+1} = y_i \quad i = 1, 2, \cdots n$. The first system of equations can be efficiently solved by Gaussian elimination.

# B  Experiments

In this section, we describe our experimental setup and present additional analysis experiments to evaluate the functioning of SURM. Our code is available at https://github.com/arijitthegame/structured-matrices-PEFT.

## Contents

## B.1  Hyperparameters

In this section, we provide the details of the hyperparameters used in our experiments. For GLUE tasks, we use the LORA hyperparameters that are used in the original LoRA paper except we use $r = 1$ to parameter match our methods as well as $\alpha = 1$.

For all the experiments, we use AdamW optimizer [47] with a warmup ratio of 0.06, a linear learning rate scheduler, and a sequence length of 128. For our methods and the Compacter baseline, we use a batch size of 64. We report the rest of the hyperparameters in Table 4. The code to run NLP experiments is developed using PyTorch using Huggingface, Adapter-transformer, PEFT libraries, and the original LoRA codebase. For ViT experiments, we use JaX [5] and the open-sourced JAX implementation of ViT.

**Additional Details on the Pinwheel Experiment**. First, we provide a figure of the pinwheel dataset used to showcase the approximation qualities to LDRMs (see Fig 9 left). We provide additional details on the pinwheel experiment. We tried out 2 settings : **(a)** simple neural network training for 2000 epochs, **(b)** the embedding (bottom) layer is frozen and the rest of the network is trained for 2000 epochs. This can be thought of fitting a feature extractor on top of a randomized projection. The setting **(b)** is presented in the main paper while setting **(a)** is presented in Appendix B.2. Next, we provide additional details for our text and vision experiments.

Table 4: Hyperparameters used for our GLUE experiments

| Method | Hyperparameters | RTE | MRPC | QNLI | QQP | SST-2 | MNLI | STSB | COLA |
|---|---|---|---|---|---|---|---|---|---|
| LoRA | Batch Size | 32 | 16 | 32 | 16 | 16 | 16 | 16 | 32 |
| | # Epochs | 80 | 30 | 25 | 25 | 60 | 30 | 40 | 80 |
| | Learning Rate | 5e-4 | 4e-4 | 4e-4 | 5e-4 | 5e-4 | 5e-4 | 4e-4 | 4e-4 |
| Kronecker-LoRA | Weight Decay | 0.0 | 0.25 | 0.1 | 0.1 | 0.1 | 1e-3 | 0.25 | 0.1 |
| | # Epochs | 60 | 70 | 60 | 80 | 60 | 80 | 70 | 70 |
| | Learning Rate | 7e-4 | 2e-3 | 2e-3 | 2e-3 | 2e-3 | 2e-3 | 2e-3 | 2e-3 |
| | Dropout | 0.1 | 0.1 | 0.1 | 0.1 | 0.1 | 0.1 | 0.15 | 0.1 |
| Circulant-LoRA | Weight Decay | .25 | .15 | .1 | .1 | .1 | .1 | .25 | .1 |
| | # Epochs | 70 | 60 | 80 | 80 | 60 | 80 | 80 | 70 |
| | Learning Rate | 2e-3 | 2e-3 | 2e-3 | 2e-3 | 2e-3 | 2e-3 | 2e-3 | 2e-3 |
| | Dropout | 0.15 | 0.0 | 0.1 | 0.1 | 0.1 | 0.1 | 0.1 | 0.0 |
| Toeplitz-LoRA | Weight Decay | 0.0 | 0.0 | 0.0 | 0.0 | 0.0 | 0.0 | 0.0 | 0.0 |
| | # Epochs | 70 | 60 | 80 | 80 | 60 | 80 | 70 | 60 |
| | Learning Rate | 7e-4 | 5e-4 | 7e-4 | 7e-4 | 7e-4 | 7e-4 | 7e-4 | 7e-4 |
| | Dropout | 0.1 | 0.1 | 0.1 | 0.1 | 0.1 | 0.1 | 0.1 | 0.1 |
| Kronecker-Adapter | Weight Decay | 0.2 | 0.2 | 0.2 | 0.2 | 0.2 | 0.2 | 0.2 | 0.2 |
| | # Epochs | 70 | 70 | 80 | 80 | 60 | 80 | 70 | 60 |
| | Learning Rate | 2e-3 | 2e-3 | 3e-3 | 3e-3 | 3e-3 | 3e-3 | 3e-3 | 3e-3 |
| | Dropout | 0.1 | 0.1 | 0.1 | 0.1 | 0.1 | 0.1 | 0.1 | 0.1 |
| Circulant-Adapter | Weight Decay | 1e-4 | 0.0 | 1e-4 | 1e-4 | 1e-4 | 1e-4 | 0.2 | 0.0 |
| | # Epochs | 70 | 70 | 80 | 80 | 60 | 80 | 70 | 60 |
| | Learning Rate | 2e-3 | 2e-3 | 2e-3 | 2e-3 | 2e-3 | 2e-3 | 3e-3 | 2e-3 |
| | Dropout | 0.1 | 0.1 | 0.1 | 0.1 | 0.1 | 0.1 | 0.1 | 0.1 |
| Toeplitz-Adapter | Weight Decay | 0.2 | 0.2 | 0.1 | 0.1 | 0.2 | 0.1 | 0.2 | 0.2 |
| | # Epochs | 70 | 70 | 80 | 80 | 60 | 80 | 70 | 60 |
| | Learning Rate | 2e-3 | 2e-3 | 3e-3 | 3e-3 | 3e-3 | 3e-3 | 3e-3 | 3e-3 |
| | Dropout | 0.1 | 0.1 | 0.1 | 0.1 | 0.1 | 0.1 | 0.1 | 0.1 |
| Compacter | # Epochs | 70 | 70 | 80 | 80 | 60 | 80 | 70 | 60 |
| | Learning Rate | 3e-3 | 3e-3 | 3e-3 | 3e-3 | 3e-3 | 3e-3 | 3e-3 | 3e-3 |

**NLP Experiments**. We train the LoRA-BERT using PEFT library from Huggingface [78]. The hyperparameters used by the original authors are used in this setting. For experiments comparing with the LoRA baseline, we parameter match the LoRA updates with our SURMs, thus the LoRA updates are given by rank 1 matrices. we inject the LoRA modules in query, key, value projection matrices and also show ablations where we remove the adaptation from the key matrix.

For the adapter setting, we apply the GeLU non-linearity. Kronecker-based adapter, though similar to various other methods, was never tested in the BERT-setting and thus we implement it here. And in all cases, we add an (optional) dropout on the representations coming from these adaptive layers. We train the compacter baseline using the adapter-transformers library [59]. For the compacter parameters, we use $n = 4$ (number of terms in the Tucker decomposition) and the reduction factor to create the low rank matrices to be 16. All our methods have the same number of training parameters $2d$ (excluding bias terms), which gives the reader a holistic overview of how these matrices perform when injected into different PEFT paradigms. All the baseline methods use a batch size of 32, whereas our methods use a batch size of 64. AdamW [47] optimizer is used for all experiments.

**Image Classification Experiments**. For the image experiments, we use Adam optimizer [33] with 20k max iterations per dataset with a batch size of 64. The learning rate used is 5e-5 except for SVHN where we use a learning rate of 5e-4. The experiments are run on TPUv4 $4 \times 2$ compute resources.

**Large Scale Experiments**. In this experiment, we investigate if large ranks are needed for learning new tasks. To circumvent the pre-trained knowledge in the Transformers, following [29], we generate random 10K pairs of Universally Unique Identifiers (UUIDs), each pair comprising two UUIDs with 32 hexadecimal values. The task requires the LLM to generate the corresponding UUID based on the input UUID. We use LLaMA-2 7B as base model [69] for this experiment. For the LoRA setting, we apply rank 256 matrices to *only* the linear layers in the attention layer. For MoRA, we use the same setting as in [29]. However, note that in [29], the authors apply adaptation to all linear layers.

Table 5: Performance of SURM and other baselines on GLUE benchmark. We report the MCC score CoLA, F1 score for MRPC, Spearman correlation for STSB, and accuracy scores for the other tasks. All results are obtained by averaging over 3 seeds. Best numbers are highlighted in **bold** and the second best numbers is underline.

| Method | # Params $(\times 10^6)$ | RTE 2.5k | MRPC 3.7k | QNLI 105k | QQP 364k | SST-2 67k | MNLI 393k | STSB 7k | COLA 8.5k |
|---|---|---|---|---|---|---|---|---|---|
| BERT-baseline [20] | 110 | 66.2 | 90.5 | 91.3 | **91.4** | 92.6 | 84.1 | 88.8 | 59.5 |
| Adapter (Houlsby) [59] | 1.8 | 69.8 | 91.5 | 91.2 | 90.8 | 92.8 | 84.1 | **89.2** | 59.1 |
| Adapter (Pfeiffer) [59] | 0.9 | 70.8 | 89.7 | 91.3 | 90.5 | 92.2 | 84.1 | 89.0 | 58.9 |
| *AA* [53] | 0.7 | 64.25 | 85.09 | 89.96 | 88.09 | 91.31 | 82.89 | 88.25 | 51.44 |
| BitFit [84] | 0.1 | 72.3 | 90.4 | 90.2 | 85.6 | 92.1 | 81.4 | **89.2** | 58.8 |
| Compacter [51] | 0.11 | 72.84 | 90.18 | 91.08 | 90.6 | 92.1 | 83.26 | 88.64 | 59.6 |
| LORA [26] | **0.06** | 71.12 | 90.43 | 90.45 | 90.1 | 92.66 | 83.06 | 88.69 | 57.83 |
| Prefix [43] | 0.19 | 70.54 | 89.93 | 90.76 | 89.12 | 91.93 | 82.78 | 85.93 | 58.86 |
| Serial [90] | 0.89 | 68.01 | 88.65 | 91.06 | 90.52 | 91.93 | 84.18 | 84.75 | 59.73 |
| AdaMix [76] | 0.89 | 70.11 | 90.91 | **91.52** | 90.22 | 92.06 | 84.25 | 86.86 | 59.11 |
| UniPELT [52] | 1.38 | 67.07 | 88.72 | 91.09 | 90.69 | 92.52 | **84.28** | 84.22 | 60.13 |
| Parallel [90] | 7.67 | 68.52 | 90.72 | 90.83 | 90.74 | 92.13 | 73.93 | 86.52 | 58.72 |
| MAM [23] | 7.67 | 69.10 | 91.46 | 90.85 | 90.76 | 83.94 | 83.31 | 89.01 | 47.87 |
| AUTOPEFT [90] | 1.54 | 72.35 | 91.5 | 91.12 | 90.64 | 92.22 | 84.01 | 89.17 | **60.92** |
| SURM (*Kronecker-Adapter*) | **0.06** | **72.96** | 91.11 | 90.53 | 89.86 | 92.66 | 83.01 | 88.94 | 58.77 |
| SURM (*Toeplitz-Adapter*) | **0.06** | 72.92 | 91.08 | 90.47 | 89.54 | 92.55 | 83.04 | 89.08 | 59.56 |
| SURM (*Circulant-Adapter*) | **0.06** | 72.12 | **91.55** | 91.24 | 89.97 | **93.0** | 83.45 | 88.78 | 59.2 |
| SURM (*Kronecker-LoRA*) | **0.06** | 71.35 | 90.08 | 90.87 | 90.0 | 92.78 | 83.02 | 88.91 | 60.35 |
| SURM (*Toeplitz-LoRA*) | **0.06** | 71.4 | 90.96 | 90.56 | 89.95 | 92.4 | 82.54 | 88.74 | 58.83 |
| SURM (*Circulant-LoRA*) | **0.06** | 71.84 | 91.02 | 90.64 | 90.15 | 92.68 | 82.87 | 89.18 | 59.97 |

For the SURM methods, we follow the same setting as applying the adaptation to only the attention layers. The number of factors of $\mathbf{W}(\mathbf{G}, \mathbf{H})$ is chosen to be 4 (i.e. same number of parameters as a rank 4 matrix). This experiment also highlights that for effective transfer learning LoRA needs to be applied to *all* linear layers (which is well-known in the LLM community). In Fig. 8, we observe that LoRA and MoRA struggles to fit the data whereas our method converges.

**Image Segmentation Experiments**. For this experiment, we use Synapse multi-organ segmentation dataset. 30 abdominal CT scans in the MICCAI 2015 Multi-Atlas Abdomen Labeling Challenge are divided into 18 training cases and 12 test cases. There are 3779 axial contrast-enhanced abdominal CT images in total and the training set contains 2212 axial slices. All the CT volumes contain $85 \sim 198$ slices and each slice includes $512 \times 512$ pixels with a spatial resolution of ($[0.54 \sim 0.54] \times [0.98 \sim 0.98] \times [2.5 \times 5.0]$mm$^3$). We use the Segment-Anything-Model (SAM) [34] as the foundation model for this task. There has been a number of works in adapting various PEFT methods to fine tuning SAM. We follow the training details in [86]. More specifically, we adapt the $\mathbf{Q}, \mathbf{V}$ in ViT-B image encoder in the SAM and normally finetune the small decoder head. Finally, in this small data regime, we use the Circulant variant as it is our most performant variant in this case (see Fig. 6). We report the Dice similarity coefficient (DSC) metric for each of the 8 organ segmentation as well as the average DSC score for all (higher is better). The SAMed model uses a LoRA rank 4 in $\mathbf{Q}, \mathbf{V}$. For a fair comparison, we include LoRA rank 1, matching the *exact* parameter count of Circulant. We use an A100 40GB GPU for this experiment.

## B.2 Additional Experiments

In this section, we provide additional experiments to showcase the efficacy of SURMs.

**Experiments on Large Scale Data** Next, we conduct additional experiments on the ImageNet-1k dataset [18]. The goal is to show how our methods can scale up to extremely large datasets. We observe that SURM achieves comparable performance to other PEFT methods and even achieving comparable performance to the full fine-tuning results (see Table 6).

We further evaluate the performance of SURM in a large-scale setting using the iNat2021 dataset [72], which contains over $2.7$ million training images, **100K** validation images, and **500K** test images, spanning $10,000$ species (classes). Fine-tuning a ViT model on this dataset achieves an accuracy of 69.98%, while SURM (circulant) achieves 69.01%. Notably, our method requires only **55K**

Table 6: Comparison of the performance of SURM and baseline PEFT methods on ImageNet-1k

| | SURM (*Kronecker*) | SURM (*Toeplitz*) | SURM (*Circulant*) | Linear Prob. | VPT-Shallow | VPT-Deep | Fine-tuning | SSF |
|---|---|---|---|---|---|---|---|---|
| # Params (M) | 0.055 | 0.055 | 0.055 | 0.049 | 0.063 | 0.531 | 86.63 | 0.240 |
| Accuracy | 83.14 | 80.17 | 82.67 | 82.04 | 82.08 | 82.45 | **84.1** | 83.10 |

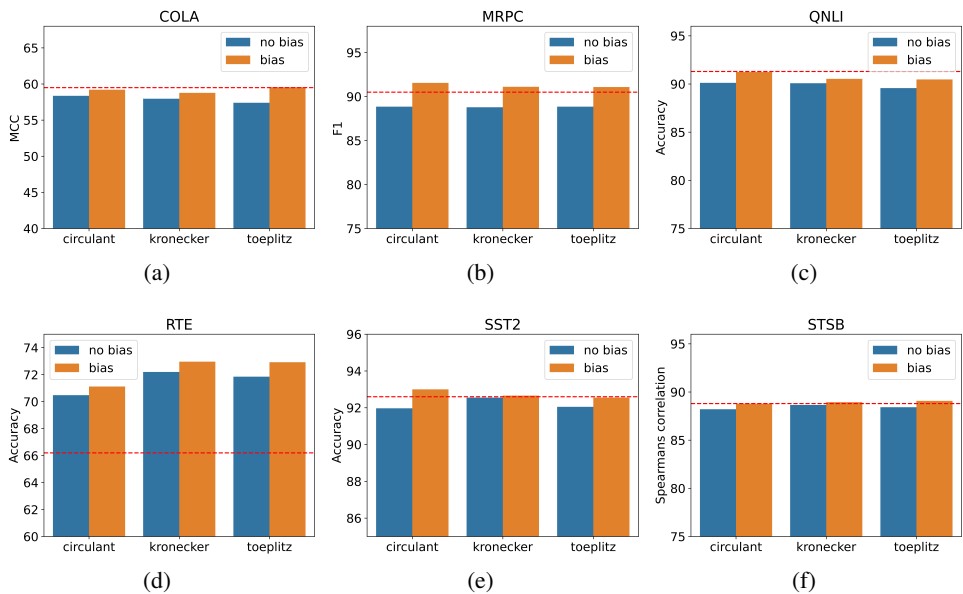

Figure 10: Figures showing the effect of using bias on various adapters on the GLUE tasks. The red dashed line is the full fine-tuning baseline which is almost **2000x** larger than our adapters.

parameters, compared to $86M$ for full fine-tuning, demonstrating its efficiency in parameter usage while maintaining comparable accuracy.

**Ablation Experiments**. Here, we show the effect of various design choices. Figure 10 illustrates the impact of incorporating the bias term in our adapters. Bias term provides a boost across all tasks and the adapters, the boost being smaller on the Kronecker adapter. Without the bias terms, the sizes of the adapters are around **.04M**, providing an even lightweight but still capable method. Therefore, if there are concerns regarding storage and latency, opting for adapters without bias is a viable option. Moreover, we show the effect on only adapting $\mathbf{Q}, \mathbf{V}$ instead of $\mathbf{Q}, \mathbf{K}, \mathbf{V}$ as shown in the main paper. Table 7 shows that on GLUE tasks, there is a minimal effect for not adapting the $\mathbf{K}$ matrix.

### B.3 Comparison of SURM Kronecker Adaptations with Baselines

As mentioned earlier, adaptation using Kronecker product is not new and has been investigated in several works [51, 21, 24]. In both [51] and [24], the authors use the Kronecker decomposition of the weight matrix (in the first case, the weight matrix belongs to an adapter layer and in the second case the weight matrix refers to updates as in the case of LoRA). Write $\mathbf{W} = \sum_{i=1}^{n} \mathbf{A_i} \otimes \mathbf{B_i}$. Furthermore, the authors assume that $\mathbf{B_i}$ is low rank and can be written as $\mathbf{B_i} := \mathbf{u}_{ij}\mathbf{v}_{ij}^{\top}$. The weights $\mathbf{A_i}$ can also be assumed to low weights or be shared among various layers leading to substantial efficiency gains. Our method is a simplified version of the above where $n = 1$. Other main difference between the above methods and ours are : the matrices considered in the above works are square matrices whereas they are almost never square unless the dimension of the transformer is a perfect square and is set up such that the number of parameters are reduced while the rank is as high as possible, contrary to the above. Similar considerations of low rank factors in tensor decomposition are also used in [30]. Our Kronecker adaptation is same as that of [21] in the LoRA setting. In the adapter setting, our implementation follows closely the Houlsby architecture and is a little different than that of [21]. Thus, we implement the Kronecker adaptation in *both* LoRA and adapter settings and showcase its

Table 7: LoRA ablation experiments on GLUE benchmarks. MCC score is reported for CoLA, F1 score is reported for MRPC, and Spearman correlation is reported for STSB. Accuracy scores are reported for the other tasks. All results are obtained by averaging over 3 seeds. The best results are in **bold** and the second best results are underlined.

| | # Params ($\times 10^6$) | RTE | MRPC | QNLI | QQP | SST-2 | MNLI | STSB | COLA |
|---|---|---|---|---|---|---|---|---|---|
| Bert-baseline [20] | 110 | 66.2 | 90.5 | **91.3** | **91.4** | **92.6** | **84.1** | 88.8 | 59.5 |
| LoRA [26] | 0.04 | 70.76 | 89.02 | 89.4 | 89.27 | 92.2 | 80.27 | **88.89** | 59.08 |
| SURM (*Kronecker*) | 0.04 | 70.04 | 89.06 | 90.54 | 89.35 | 91.74 | 80.41 | 88.74 | **59.6** |
| SURM (*Toeplitz*) | 0.04 | **72.56** | **91.04** | 89.65 | 89.67 | 92.14 | 80.93 | 88.77 | 58.05 |
| SURM (*Circulant*) | 0.04 | 71.14 | 90.48 | 89.91 | 89.83 | 92.2 | 80.6 | 88.76 | 59.16 |

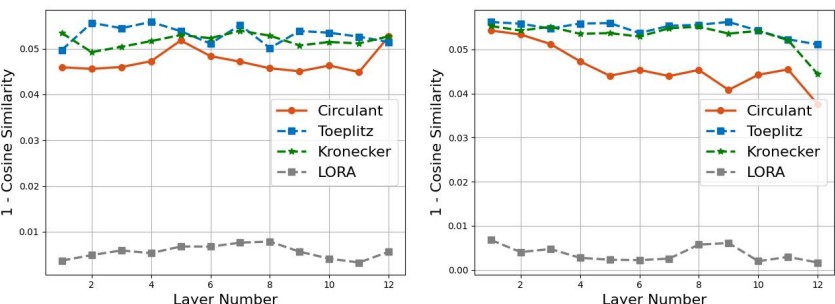

Figure 11: **Left**: Cosine similarity between the query matrices and **Right**: cosine similarity between value matrices for the BERT model on MRPC dataset.

versatility across both vision and language. Moreover, we present this approach as an example of a principled approach to tackle the problem of PEFT.

## B.4    Analysis of Weight Matrices in Fine-tuned Models

In this section, we analyze the weights of various fine-tuned models. Even though prior works have found the updates of the weight matrices to have low intrinsic dimension [1] (ID), the updates themselves are of high rank. This is confirmed by looking at the fine-tuned BERT models on various GLUE tasks as well as ViT models fine-tuned on CIFAR10, CIFAR100, and ImageNet. Moreover, we simulate a high-rank LoRA setting on GLUE where we freeze all the weights excepts except for $\mathbf{Q}, \mathbf{K}, \mathbf{V}$. In that scenario, we manage to replicate the full fine-tuning performance using fewer training epochs than that of LoRA. A quick analysis of the updates reveal that they have *full* rank.

Many works have delved into intrinsic dimensionality for well-known image classification datasets [60]. These works show that the images have low intrinsic dimensionality compared to the pixel spaces but the dimensionality increases when augmentations like Gaussian noise is added. Recent work [49] studies the intrinsic dimensions of various self-supervised image models. Comparing their results with that of fully supervised ViT models, we observe that the self-supervised models exhibit slightly higher IDs. This is not surprising as the SSL encourages the representations to be spread over an unit hyper sphere. Thus, we believe that various low rank adaptations may fail in situations where the IDs might be high (in case of OOD data) [75].

Encouraged by this analysis, we next investigate the trained weights emerging from our methods. We observe that they have **high rank** across all vision and text tasks and various fine-tuning strategies. The largest possible rank of the Kronecker matrices considered in this work is **576** and all of our trained matrices are of rank **576**. For rational circulant matrices $\mathbf{C}$, the non-singularity of such matrices is related to divisibility by cyclotomic polynomials. More generally, if we denote by $\mathbf{c} = (c_0, .. c_{n-1})$ the first column of $\mathbf{C}$, then:

$$\det(\mathbf{C}) = \prod_{j=0}^{n-1} \left( c_0 + c_1 \omega_j + c_2 \omega_j^2 + \cdots c_{n-1} \omega_j^{n-1} \right),$$

Table 8: CKA between full finetuned weights and the SURM weights

|  | LoRA | Circulant | Symmetric Toeplitz | Toeplitz |
|---|---|---|---|---|
| CKA | 0.014 | **0.1821** | 0.1343 | 0.1618 |

where $\omega_j = e^{\frac{2\pi \mathbf{i} j}{n}}$ and $\mathbf{i}^2 = -1$ (for more details see [14]). This fact allows us to efficiently test for the non-singularity of the circulant matrices. In all our cases, we found our matrices to be non-singular. Regarding Toeplitz matrices, there is a large body of literature that discusses the inversion of such matrices (see Appendix A.8). Using the methods discussed above, we find that the Toeplitz adaptations are invertible, thus full-rank. Therefore, we hypothesize that the high rank compensates for the deficiency of training parameters.

To further explore the differences between the parameters learned by LoRA vs. that learned by SURM methods we performed another set of experiments. We calculate the cosine similarity between the weights learned by the PEFT methods ($\hat{\mathbf{W}} = \mathbf{W} + \alpha \Delta \mathbf{W}$) and $\mathbf{W}$ (pre-trained weights). A smaller cosine similarity would tell us that SURMs help us in exploring parameters further away from the pre-trained weights ($\mathbf{W}$).

We test our hypothesis on the BERT model finetuned on the MRPC dataset by SURMas well as by LoRA. We report the (1-cosine similarity($\hat{\mathbf{W}}, \mathbf{W}$)) for both query and key across multiple layers (see Fig 11). We see that LoRA-learnt weights are very similar to the pretrained weights whereas SURMs explore a larger space (as shown by higher dissimilarity). This observation is not too dissimilar to that of [91].

**Analysis of trained weight matrices for Pinwheel data**. We also want to answer the question: **Q:** *How similar are the representations learned by networks with the SURM layers compared to the full finetuned networks?*

We evaluate the CKA similarity [38] between the full fine-tuned network and the network with the LDR layers. CKA is a widely used metric to compare representations coming from different neural networks. We observe that LDR networks have higher CKA similarity with fully finetuned networks than their LoRA counterparts.

## B.5 Guidance for Practitioners

To translate our framework into actionable insights, we aim to highlight several key properties of the various classes of SURMs that help us in making the final recommendation. In all our experiments, we found that on average that **circulant** variant achieves the largest number of best performances across multiple datasets (Figure 1, Table 1, 2, 3). Moreover, in the low data regime, it is clear that the circulant is the most *performant* variant as well (Figure 6).

The time complexity of LDR matrices is sub-quadratic, in particular, time complexity for both: the circulant and the Toeplitz variant is the same but the Toeplitz one is slower by a factor of 2. The gradients allow for a very simple formula, which is computed in sub-quadratic time (see eq 14 in [15] for the circulant matrix and Proposition 3.6 in [66] for the Toeplitz matrix). Therefore, our general recommendation to practitioners is to use the circulant variant of SURM. It is relatively fast and our most accurate variant.

## C  Broader Impact & Limitations

Fine-tuning large pre-trained Transformers for downstream tasks requires substantial computational resources. We hope that this work addresses this important problem by reducing the overall computational budget while maintaining high accuracy. We believe that SURMs will make Transformers accessible to researchers and academics world wide and also reduce the carbon footprint associated with training these models. While democratizing powerful Transformers' technologies with those methods, one must still be cautious of the potential harmful biases, inherent to models pre-trained on the internet-scale data. One of the main limitations is the absence of custom kernels for our methods. Despite their theoretical speed advantage, popular methods like LoRA have been extensively optimized by the machine learning community for efficient execution on hardware.

