# OpenReview forum: "Structured Unrestricted-Rank Matrices for Parameter Efficient Finetuning"
_NeurIPS.cc/2024/Conference — NeurIPS 2024 poster_

### Official Review · Reviewer_Kv7A · 2024-07-09

**Soundness:** 2
**Presentation:** 2
**Contribution:** 3
**Rating:** 5
**Confidence:** 4

**Summary:**

The paper explores the use of structured matrices instead of low-rank matrices for approximating the finetuning updates for Transformer models. The sturcture imposed on the approximation matrices determines the number of trainable parameters. The paper explores use of two structured matrices: Circulant and Toeplitz matrices, and Kronecker product of Matrices for finetuning language models and vision models. The paper shows that structured matrices are better at approximating various class of matrices like random (full rank), near-low-rank and near-low-intrinsic-rank than low-rank matrices, and this translates to better performance when using them for finetuning at a lower parameter cost.

**Strengths:**

- Use of structured matrices for finetuning achieves on-par/better performance than prior works at a lower parameter cost.
- The toy experiments show that structured matrices can approximate matrices better than low-rank matrices (controlling for the number of parameters). The authors verifiy this through two experiments:
    - Approximating Symmetric Positive Definite Matrices
    - Fitting a toy dataset (pinwheel with Gaussian noise) using a neural net with layers composed of structured and low rank matrices.

**Weaknesses:**

1. **No studies on more difficult tasks**:
- While I understand that the authors have shown the effectiveness of the proposed method on smaller models, it would be interesting to see how the proposed method performs on larger models for more difficult tasks like language generation (for e.g., insturction tuning) or math/commonsense reasoning which are commonly used to study PEFT methods on large language models.
2. **Lack of details on choice of hyperparameters**:
- The paper does not provide sufficient details about how the data is split, if all the methods are compared by training and evaluating on the same data splits.
- Moreover, the paper does not provide details on how the hyperparameters are chosen, but only the final values are mentioned. (Section E in the appendix.)
3. **Minor presentation details**:
- Figure 1: caption says the presented methods are in top left (in green), but appear to be in the top right in the plots, and also do not apprear to be in green.

**Questions:**

1. **Section 4.1**:
- What is the significance of Section 4.1? What does the comparison between Circulant and Toeplitz matrices with W(G,H) convey? Could the authors provide more explanation about why this comparison is done, especially since the final PEFT algorithm uses only Circulant and Toeplitz matrices?
- Do all the methods in this study approximate the exact same matrix? Why do the initial errors vary by such a large margin across W(G,H), and Circular and Toeplits matrices?
- Why do Circulant and Toeplitz matrices use more iterations than W(G,H) (200 vs. 2000)? If the number of iterations is the same (i.e., W(G,H) is optimized for 2000 iterations), do the results change?
- Why is there no study for approximation of random matrix using Toeplitz Circular matrices?
- Why is low-rank(I=20) + epsilon not considered for Circular and Toeplitz matrices?

2. **Experiments on image classification using ViTs**:
- What is the pretraining data and task for the base ViT?
- How is the data split into train, validation and test splits, and on which split is are the results reported?
- How and on which split are the hyperparameters tuned? Section E, lines 699-700 mention that the experiments use a learning rate of 5e-5, but SVHN uses 5e-4. How is this chosen?

3. **Experiments on GLUE**:
- What is the difference bewteen SURM (Circular) and SURM (Circular-LoRA/Adapter) (similarly Toeplitz and Kronecker)?

**Limitations:**

Authors have addressed the limitations.

---

> ### Author Rebuttal · Authors · 2024-08-06
>
> We would like to sincerely thank the Reviewer for their very valuable feedback and comments.
>
> > Studies on larger tasks:
>
> One of the key things that we showed in our experiments was the improved performance in the low data regime. Specifically across a plethora of low-resource image experiments (VTAB-1K) we have shown the improved performance of SURMs (circulant). The NLP experiment on Glue was provided to showcase the ability of SURMs to extend to other modalities (text) and another class of small parameter models (adapters). We will make it more clear in the paper.
>
> > Choice of hyperparameters:
>
> Datasplit: For vit-image experiments (table 1) we used the standard dataset split in tensorflow tfds datasets (https://www.tensorflow.org/datasets). VTAB-1K provides its own split and code which we used (github). For image segmentation we used the same split as the Segment-Anything-Model (SAM) [33]. For Glue we used the standard train/dev split (https://gluebenchmark.com/).
>
> > How the hyperparameters are chosen
>
> All hyperparameters are tuned on training/val set and numbers are reported on separate test set. We will update the manuscript with this information.
>
> Thanks for pointing out the typo in the caption of Figure 1. We will fix them.
>
>
> > Significance of Section 4.1: motivation, comparisons and conclusion
>
> We apologize for the confusion. Our motivations for the two separate parts (left and right column of figure 4) are different. To save space we put these two experimental results together. Our motivation for the left column is to showcase the ability of general LDRs (defined in eq 2) to approximate various matrices i.e. random, low rank and low intrinsic rank and motivate our choice of LDRs with r=1. From the bottom two figures on the left (i.e. approximating low rank and low intrinsic rank with LDRs) we learn that LDRs (with r=1 and r=2) are great at reducing the approximation error (They are in the top 3 best approximating errors (lowest error)). This further provides motivation and empirical justification that LDRs with (r=1) may be well suited to approximate matrices with some structure.
>
> The motivation for the right column is to compare and contrast the two variants of LDRs that we proposed i.e. circulant and Toeplitz.
>
> Hence we should not be comparing the right with the left column. Please see the left and the right as its own set of experiments. In each figure, all techniques approximate the same matrix. The first set of results shown are after a single iteration hence the difference within the same figure. We can provide the left column experiments with a higher number of iterations but the objective is to discriminate among all the LDRs.
>
>
>
> > What is the pretraining data and task for the base ViT?
>
> We used the vit-base model and the clip-base model from HuggingFace (https://huggingface.co/google/vit-base-patch16-224)
>
> > How is the data split into train, validation and test splits, and on which split are the results reported?
>
> For vit-image experiments (table 1) we used the standard dataset split in tensorflow tfds datasets (https://www.tensorflow.org/datasets). VTAB-1K provides its own train/dev/test split and code for the same, which we used (github). For image segmentation, we used the same split as the Segment-Anything-Model (SAM) [33]. For Glue we used the standard train/dev split (https://gluebenchmark.com/). All numbers are reported on the test set (no hyperparameters were trained on. Thanks for pointing this out, we will update it in the manuscript
>
> > How and on which split are the hyperparameters tuned? Section E, lines 699-700 mention that the experiments use a learning rate of 5e-5, but SVHN uses 5e-4. How is this chosen?
>
> For image classification experiments we used the same setup for all datasets except for SVHN where the training accuracy didn’t move much when using 5e-5 as the learning rate. All hyperparameters have been turned on the training/dev set while reported results are on the test set.
>
>
>
> > What is the difference between SURM (Circular) and SURM (Circular-LoRA/Adapter) (similarly Toeplitz and Kronecker) used in GLUE?
>
> In the interest of saving space our detailed discussion regarding how SURMs can be used as Adapters has been moved to Appendix C. The difference between SURM(*-LoRA) and SURM(*-Adapters) is the setting under which low displacement rank matrices can be used. For example, circular-LoRa refers to using circulant matrices for lora style update ie $\hat{W} = W + \alpha \Delta W$ where $\Delta W$ is circulant (please see 5.1 for Lora style PEFT method). circular-Adapter implies using the adapter method for updating the model (please see equation 7 in section C in the Appendix). Our objective was to elaborate that both these styles of training can benefit from SURMs.
>
> [33] Customized Segment Anything Model for Medical Image Segmentation,  Zhang et al 2023.
>
> [80] A large-scale study of representation learning with the visual task adaptation benchmark, Zhai et al. 2020.

---

> > ### Comment · Reviewer_Kv7A · 2024-08-09
> >
> > I appreciate the detaild response from the authors. My questions and weakness about hyperparameter tuning strategy have been  addressed. However, I am still not convinced with the authors' reasoning about large scale experiments. Even in settings where data is scarce, when applied to larger models in the scale of billion parameters and to more complex tasks like language generation, methods are not guaranteed to work as well. Without these experiments, it's hard to judge the effectiveness of the method. Hence, I will maintain my score.

---

> ### Author Response · Authors · 2024-08-11
>
> > Large data Regime
>
> We investigate the performance of SURM in a large data regime using the iNat2021 [1] dataset. iNat2021 has over **2.7 million** training images, 100K validation images, and 500K test images, representing a wide array of **10,000 species (classes)**.
>
> Full fine-tuning: **69.98%** vs SURM (circulant): **69.01%**
>
> We observe that SURM achieves similar results to full-finetuning using only **55K** parameters as opposed to **86M** in full fine-tuning. We will add these details to the manuscript.
>
> [1] Benchmarking Representation Learning for Natural World Image Collections, Horn et al. 2021..
>
>  > Large Model regime/Small data Regime
>
> Thank you for the excellent comment. We present an example below where *methods like LoRA and MoRA struggle to learn a complex task in a low data regime with a large-scale model.*
>
> Following the suggestion reviewer o41z, we created 10k pairs of UUIDs and tested the memorization capability of LLMs. We use the large-scale **Llama-2-7B** model [2]. The goal of this experiment is to show that models struggle to learn out-of-distribution data when using low-rank updates. Since the pdf can not be updated, the results are presented below:
>
> | Method &#8595; / Steps &#8594;    | 25   | 100  | 200  | 300  | 400  | 500  | 600   | 700   | 800   | 900   | 1000  | 3000 |
> |----------------------------------|------|------|------|------|------|------|-------|-------|-------|-------|-------|------|
> | LoRA (2.9% Param)                | 2.42 | 2.29 | 2.30 | 2.29 | 2.28 | 2.28 | 2.28  | 2.28  | 2.29  | 2.27  | 2.29  | 2.29 |
> | MoRA (2.9% Param)                | 2.43 | 2.29 | 2.30 | 2.28 | 2.28 | 2.28 | 2.29  | 2.28  | 2.29  | 2.27  | 2.29  | 2.26 |
> | Circulant (0.01% Param)          | 3.40 | 3.32 | 2.96 | 2.70 | 2.35 | 2.04 | 1.92  | 1.83  | 1.78  | 1.74  | 1.72  | 0.97 |
> | Circulant+Skew-circulant (0.04%) | 3.78 | 0.34 | 0.09 | 0.05 | 0.02 | 0.02 | 0.006 | 0.009 | 0.004 | 0.003 | 0.001 | 0.0  |
>
>
> We observe that LoRA and MoRA struggle to fit the data (cross-entropy loss around 2.3) whereas our circulant variant achieves a loss of **0.97**. In this experiment, we used a high rank=256 for both LoRA and MoRA, and modified the $Q, K, V$ parameters for all methods.
>
>
> Furthermore, we show the effect of increasing the number of training parameters by using sums of products of circulant and skew-circulant matrices. A skew circulant matrix $S = (s\_{jk})\_{j,k=0}^{n-1}$ is said to be skew-circulant if $s\_{jk} = s\_{j−k}$ and $s\_{−l} = −s\_{n−l}$ for $1 \leq l \leq n − 1$. The motivation for using this particular sum of products comes from the approximation quality of such matrices (see Theorem 1 in [3]). This is evident in practice as circulant+skew-circulant variant obtains a loss=**0** and converges much faster.
>
> This result and our toy experiment (Figure 5) consistently show that low rank updates may struggle to fit various data regimes and that unrestricted-rank matrices may be required to alleviate this issue. SURMs solve this problem using structured matrices (keeping the parameter budget low) while allowing for arbitrary ranks.
>
> We will add this result to the manuscript.
>
> [2] MoRA: High-Rank Updating for Parameter-Efficient Fine-Tuning Jiang et al. 2024
>
> [3] Structured Transforms for Small-Footprint Deep Learning Sindhwani et al. 2015

---

> > ### Comment · Reviewer_Kv7A · 2024-08-12
> >
> > I appreciate the authors' efforts on experiments with Llama 2 7B. However, I'm suspicious about the experimental settings used for the latest experiments on UUID memorization. From the original MoRA paper [1], LoRA and MoRA can indeed memorize UUIDs well, as evidenced from the 100% character level accuracy (Table 2) in [1]. However, the authors' experiments do not reflect this. Furthermore, why have the authors reported cross-entropy instead of accuracy as done in the original paper?
> >
> > This discrepancy looks suspicious, and warrants more careful experimentation. I would encourage the authors to investigate this in detail. Given these results, I'm afraid I cannot change my score.
> >
> > [1] MoRA: High-Rank Updating for Parameter-Efficient Fine-Tuning Jiang et al. 2024

---

> > > ### Author Response · Authors · 2024-08-14
> > >
> > > Dear Reviewer Kv7A,
> > >
> > > Thank you for the excellent question.
> > >
> > > The difference between our results and that reported in MoRA [1] arises because the authors in [1] use LoRA in **all** linear layers whereas we only use it in $Q, K, V$. We aim to show that in this low-parameter regime, LoRA and MoRA struggle to fit the data whereas SURM excels in this task.
> > >
> > > The cross-entropy training loss is a valid metric for this task and was reported in Fig. 2 in the original paper [1]. We found that the generation quality depends quite a lot on the hyperparameters (which are not open-sourced yet) and couldn’t exactly replicate the results presented in [1]. Therefore, for a fair comparison, we report the convergence of the training loss for all methods.
> > >
> > >
> > > [1] MoRA: High-Rank Updating for Parameter-Efficient Fine-Tuning Jiang et al. 2024

---

### Official Review · Reviewer_o41Z · 2024-07-11

**Soundness:** 3
**Presentation:** 3
**Contribution:** 2
**Rating:** 6
**Confidence:** 3

**Summary:**

The paper introduces a new technique called SURMs which aim to use structured matrices for PEFT. The technique is tested against many different adapter variants in Vision Adaptation and Natural Language. The unique structure of the matrices allow for efficient computation of the products

**Strengths:**

- The idea of structured matrices compared to low rank ones as avenue of exploration is interesting.
- Many different adapters are used during evaluation
- The figures are clear well thought out

**Weaknesses:**

The bar for a new variant of LoRA to be substantial contribution to me is fairly high, just due to how crowded the space is.

Structured matrices for low rank adaptation has previously, as noted in the paper Knoecker products have been in previous literature, while Circulant and Toeplitz structures are, to my knowledge, have not. Nevertheless, Structured matrix adaptation has been explored in several prior papers.

The change in structure of the matrix or rank alone to me is not sufficient unless there is substantial evidence it provides benefit, both empirically and intuitively/theoretically.

The intuition and/or theory as why we might expect this class of matrices to be a useful, in terms of improved adaptation, is lacking (beyond the FFT-based efficiency). As stated previously, the space is fairly crowded for this line of research and every new modification cannot be considered significant without substantial evidence and careful reasoning behind exploration.

The empirical evidence for this change also needs to be improved upon.
PEFT experiments need to largely be conducted in regimes where full-finetuning represents a soft upper bound on the achievable performance. The most common use case for PEFT techniques is improving finetuning compute/memory efficiency, while retaining as much performance as possible. In contrast, for regimes where we don't have sufficient data for full fine-tuning to perform best, it seems that we mainly end up measuring the ability of the technique to reduce overfitting, rather than the techniques ability to retain the learning capacity of the original model at lower compute levels. This is still useful, of course, but there are many other techniques to do this outside of PEFT, and it misses evaluation for the primary use case of PEFT.

As a reference, I think the [MoRa](https://arxiv.org/pdf/2405.12130) paper does a decent job at this. The original LoRA paper doesn't do a great job with this and oversells, as the community knows from countless experiments with it. To effectively prove out which techniques hold up best its very important to conduct detailed evaluations at scale with models which can achieve SoTA on a particular dataset.

The following would make the paper much stronger:
- Stronger intuitive/theoretical justification for why we might expect this class of adaptations to perform better (with some small scale experiments or ablations proving the intuition)

- Experiments with models in NLP and Vision, where we full finetuning generally performs best (the UUID memorization from MoRa is a good example). Imagenet seems like a good candidate assuming techniques have not pertained on that data. Larger NLP datasets seem like another good candidate (extra AR pretraining, adaptation to larger downstream tasks).

From my perspective, if I actually wanted to train on the the many of the small vision datasets for example, there are many different alternate fine-tuning techniques or different models I can use to improve performance for data-starved cases, especially since those cited perform worse than other fine-tuning techniques on these datasets. (Cifar-100, DTD, SUN for example, even using the same model arches).

The paper needs to do a more careful job of picking SoTA finetunings on larger datasets and using PEFT on those.

**Only PEFT can really target the case where I have large amount of fine-tuning data, but not necessarily the compute/memory to target it with a very large model well, so this regime should be emphasized in the experiments.**

**Questions:**

What are the ranks of the other adapters used?

Why do you only test against rank 1 lora in Table 3?

What how does the techniques performance change with rank?

Why might we expect these structures to perform better than other structured matrices?

**Limitations:**

The paper should address limitations more directly

---

> ### Author Rebuttal · Authors · 2024-08-06
>
> We would like to sincerely thank the reviewer for their valuable feedback and comments.
>
> > Structured matrices for low rank adaptation has previously, as noted in the paper Knoecker products have been in previous literature, while Circulant and Toeplitz structures are, to my knowledge, have not. Nevertheless, Structured matrix adaptation has been explored in several prior papers.
>
> To the best of our knowledge, the entire class of low displacement rank matrices (LDRMs), a subclass of Structured Unrestricted-Rank Matrices (SURMs) have not been explored in the context of PEFT. This includes instantiation of LDRMs including Circulant and Toeplitz matrices. As described in line 131 in section 3, Kronecker is not necessarily an LDRM but admits to efficient matrix-vector multiplication and does fall under the general umbrella of SURMs. The key novelty is our proposal of LDR-SURMs as general approximators that are not restricted to low rank. This is further supported by our experimental results.
>
>
> > Motivation / Intuition / Justification for our proposed LDRMs
>
> After intuitively noting the inability of LowRank approximation to capture higher order updates, we have explicitly (empirically) shown in Introduction Figure 1 (right)  that Toeplitz and Circulant matrices do a better job of approximating a PSD matrix as compared to LowRank matrices. *It is well known that finetuning updates are full rank*.  Our motivation is further supported by section 4.1 which clearly shows the higher approximation ability of circulant and Toeplitz matrices. Moreover in section 4.2 we have specifically focused on the approximation quality of low rank matrices vs SURM and clearly show the advantages of LDRMs.
>
> We believe that we have provided sufficient intuition and motivation (empirically) for the use of LDRMs in particular and SURMs in general.
>
> > Experimental evidence in support of LDRMs
> We would like to emphasize that we've presented extensive experimental evidence showcasing the efficacy of SURMs.
>
> First, we compared SURMs on 6 datasets and compared our methods against **12** strong baselines. There are several analyses in the small data regime.
>
> Second, we have included results for CLIP models and a comparison on the VTAB-1K [80] benchmark, which is designed to evaluate models on diverse tasks using few examples.
>
> Third, we further have applied our work on Image segmentation and shown that we are comparable with specialized architectures developed for SOTA on image segmentation.
>
> Overall, these sets of experiments demonstrate the utility of LDRMs especially in  the low data regime.
>
> > SoTa baselines for low resource setting/smaller vision datasets.
>
> We would like to point out that VTAB-1k datasets [80], which is designed to evaluate models in the low-resource regime.  We report the baseline numbers from [29] and [49], which are the current state-of-the-art. Similarly for image segmentation tasks (table 3) SAMed[33] is a dedicated SoTa model trained for image segmentation tasks. These results show that SURM-based finetuning obtains the SOTA performance on several benchmarks.
>
>
> > What are the ranks of the other adapters used?
>
> We report the adapter baseline performances from their respective papers and their rank is 48.
>
> > Why do you only test against rank 1 lora in Table 3?
>
> SAMed [33] is an adaptation of Lora with higher rank (rank 4, the details are mentioned in Appendix E)
>
> > What how does the technique's performance change with rank?
>
> Our Circulant/Toeplitz matrices are not parameterized by rank. We have explored two ways in which the number of parameters can be increased for circulant matrices. a) $ C = \sum_i a_i C_i $ where each $C_i$ are circulant and b) $ M = \prod_i M_i $ where $M_i$ are Toeplitz (line 140.) with a modest boost in performance but at the cost of speed. We will add these details in the supplementary.
>
> Interestingly we found that the circulant and Toeplitz updates to be full rank and that the Kronecker updates to the maximum possible rank (see Appendix A)
>
>
>
> > Why might we expect these structures to perform better than other structured matrices?
>
> We have motivated the use of structured matrices other than low rank by showcasing how certain structures like PSD are better approximated by LDRs. Moreover, in section 4 we have empirically shown that in multiple scenarios LDRs perform better than low rank matrices.
>
> > results on larger datasets
>
> ImageNet results are presented in Table 7 (appendix H). In that case, our method compares favorably to full finetuning where we use only a negligible fraction of training parameters.
>
> [29] Fact: Factor-tuning for lightweight adaptation on vision transformer. Jie et al 2023
>
> [33] Customized Segment Anything Model for Medical Image Segmentation,  Zhang et al 2023.
>
> [49] Towards efficient visual adaption via structural re-parameterization. Luo et al 2023
>
> [80] A large-scale study of representation learning with the visual task adaptation benchmark, Zhai et al. 2020.

---

> > ### Comment · Reviewer_o41Z · 2024-08-08
> > **Larger Scale**
> >
> > Given the new information, I raised my score to a 4.
> >
> > I feel that there needs to be more testing in higher data regimes to be accepted. Specifically for the following reasons:
> >
> > - In cases where full fine-tuning performs best, the performance of SURMS don't seem to be as strong, such as on CIFAR-100 and CIFAR-10
> >
> > - There are much stronger techniques for nearly all the datasets tested that the reported accuracy since in the low data regime, we can either use zero-shot embeddings, ICL, or finetune smaller networks rather than relying on adapter layers.
> >
> > - A high rank adapter makes most sense when the training process has to encode a lot of data, which outstrip low rank approximations learning capacity.  So not only should there be more comparison against higher rank adapters such as MoRA but there also needs to be more testing in regimes where we expect higher ranks to be most useful, UUID memorization I feel like is an easy one to test at relatively low compute.

---

> ### Author Response · Authors · 2024-08-11
>
> > Large Model regime/ UUID Experiments
>
> Thank you for suggesting the UUID experiment and revising your score. Following [1], we created 10k pairs of UUIDs and we tested the memorization capability of the **Llama-2-7b** model. the results are presented below:
>
> | Method &#8595; / Steps &#8594;    | 25   | 100  | 200  | 300  | 400  | 500  | 600   | 700   | 800   | 900   | 1000  | 3000 |
> |----------------------------------|------|------|------|------|------|------|-------|-------|-------|-------|-------|------|
> | LoRA (2.9% Param)                | 2.42 | 2.29 | 2.30 | 2.29 | 2.28 | 2.28 | 2.28  | 2.28  | 2.29  | 2.27  | 2.29  | 2.29 |
> | MoRA (2.9% Param)                | 2.43 | 2.29 | 2.30 | 2.28 | 2.28 | 2.28 | 2.29  | 2.28  | 2.29  | 2.27  | 2.29  | 2.26 |
> | Circulant (0.01% Param)          | 3.40 | 3.32 | 2.96 | 2.70 | 2.35 | 2.04 | 1.92  | 1.83  | 1.78  | 1.74  | 1.72  | 0.97 |
> | Circulant+Skew-circulant (0.04%) | 3.78 | 0.34 | 0.09 | 0.05 | 0.02 | 0.02 | 0.006 | 0.009 | 0.004 | 0.003 | 0.001 | 0.0  |
>
>
> We observe that LoRA and MoRA struggle to fit the data (cross-entropy loss around 2.3) whereas our circulant variant achieves a loss of **0.97**. In this experiment, we used a high rank=256 for both LoRA and MoRA, and modified the $Q, K, V$ parameters for all methods.
>
>
> Furthermore, we show the effect of increasing the number of training parameters by using sums of products of circulant and skew-circulant matrices. A skew circulant matrix $S = (s\_{jk})\_{j,k=0}^{n-1}$ is said to be skew-circulant if $s\_{jk} = s\_{j−k}$ and $s\_{−l} = −s\_{n−l}$ for $1 \leq l \leq n − 1$. The motivation for using this particular sum of products comes from the approximation quality of such matrices (see Theorem 1 in [2]). This is evident in practice as circulant+skew-circulant variant obtains a loss=**0** and converges much faster.
>
> This result and our toy experiment (Figure 5) consistently show that low rank updates may struggle to fit various data regimes and that unrestricted-rank matrices may be required to alleviate this issue. SURMs solve this problem using structured matrices (keeping the parameter budget low) while allowing for arbitrary ranks.
>
> We will add this result to the manuscript.
>
> [1] MoRA: High-Rank Updating for Parameter-Efficient Fine-Tuning Jiang et al. 2024
>
> [2] Structured Transforms for Small-Footprint Deep Learning Sindhwani et al. 2015
>
>
>
> > Large data Regime
>
> We investigate the performance of SURM in a large data regime using the iNat2021 [3] dataset. iNat2021 has over **2.7 million** training images, 100K validation images, and 500K test images, representing a wide array of **10,000 species (classes)**.
>
> Full fine-tuning: **69.98%** vs SURM (circulant): **69.01%**
>
> We observe that SURM achieves similar results to full-finetuning using only **55K** parameters. We will add these details to the manuscript.
>
> [3] Benchmarking Representation Learning for Natural World Image Collections, Horn et al. 2021.
>
>
> Please let us know if you have any questions.

---

> ### Comment · Reviewer_o41Z · 2024-08-11
> **Recommend Acceptance**
>
> Thank you to the authors for the follow-up experiments and the detailed conversations. I believe that with the new experiments added to the paper and the following, the paper should be accepted.
>
> 1. Please add a sentence more information on how we change the number of parameters in this formulation (Skew-Circulant etc...)
> 2. Please discuss, as a hypothesis which lends itself to future work, why it is that this method seems to perform well at both low data and high data regimes. This is a bit counterintuitive to me.
> 3. Please address the limitations of this technique a bit more in-depth and when we expect this technique to perform worse than others.
>
> With the changes already presented, and the integration of the above points. I have raised my score to a 6 due to the strong cooperation of the authors towards providing additional information, experiments etc.. and the additional strong results and recommend acceptance.

---

> > ### Author Response · Authors · 2024-08-12
> > **Thank you for your feedback**
> >
> > We sincerely thank the reviewer for their positive feedback and for revising their score.
> >
> > > How to change the number of parameters in this formulation (Skew-Circulant etc...)
> >
> > The skew-circulant matrix, like the circulant matrix, is parameterized by its first row. We consider a general update matrix given by the equation :
> > $\Delta W := \sum\_{i=0}^{k-1} A\_i B\_i$, where $A\_i$ is a circulant matrix and $B\_i$ is skew-circulant matrix. Since both $A\_i, B\_i$ are parameterized by $n$ parameters, $A\_iB\_i$ has $2n$ parameters and thus $\Delta W$ has $2nk$ parameters which is the same as rank $k$ LoRA update. We will add these details to the manuscript.
> >
> > > Hypothesis on the effectiveness of our method in various data regimes.
> >
> > We believe our main hypothesis is that LDRs provide a better approximation of general matrices because they are not limited to being low rank. We evaluated this hypothesis and found it to be well-supported by several experiments (see Figure 1, left, and Section 4). In low-data regimes, where we have to learn a data distribution from a small sample size, the flexibility of LDRs helps improve our approximation. Similarly, in large-data regimes, where the data distribution is more complex, the lack of low-rank restrictions allows us to achieve better results.
> >
> >
> > > Limitations of our technique.
> >
> > We hypothesize that in cases where the update matrix can be well approximated by a low-rank matrix, LoRA styled methods might converge faster. We will add this to our current limitation section (Appendix M).

---

### Official Review · Reviewer_GJ7U · 2024-07-12

**Soundness:** 3
**Presentation:** 3
**Contribution:** 2
**Rating:** 5
**Confidence:** 4

**Summary:**

The paper proposed a general framework for parameter-efficient fine-tuning, based on structured unrestricted-rank matrices (SURMs) to substitute LoRA or other parameter-efficient finetuning methods. Three variants of SURMs are included, named Kronecker, Toeplitz, and Circulant, based on the matrix type. The proposed SURM method generally achieves performance comparable with LoRA/adaptors with 50% or less parameters on vision datasets like CIFAR10, SUN397, and DTD. SURM can also be incorporated into adaptors.

**Strengths:**

1. The authors introduced a novel approach to substitute LoRA. The proposed SURM method provides us with a novel view of how the matrix of the LoRA can be initialized and interacted with.
2. Integration experiments with adaptors show the possibility that this method may also be applied in other modules of models, only if they have low intrinsic ranks.

**Weaknesses:**

1. The tasks include GLUE benchmarks, but the improvements are not convincing. Most of the tasks are in trade-off with other methods. Why not increase the trainable parameter up to the same level as other methods, for example, 0.9, to examine the upper bound performance? This could be part of the ablation studies, showing the scaling relationship between the performance and trainable parameters.
2. Experiments conducted in NLP should contain new architectures, such as LLaMa.

**Questions:**

Please see the weaknesses.

---

> ### Author Rebuttal · Authors · 2024-08-06
>
> We would like to sincerely thank the Reviewer for their very valuable feedback and comments.
>
> > NLP experiments:
>
> We'd like to emphasize the breadth and depth of experimental evidence we have provided for SURMs. First, we compared SURMs on 6 image classification datasets and compared our methods against **12** strong baselines. Furthermore, we have evaluated on the VTAB-1K[80] benchmark. VTAB-1K is designed for low resource settings to evaluate models on “diverse, unseen tasks with few examples”, see abstract from [80]. Across multiple low resource settings we find that circulant works better than all prior proposed methods. This is a key contribution of our work. We further have applied our work on Image segmentation and shown that SURM is comparable with specialized architectures developed for SoTa on image segmentation. Thus our experiments demonstrate the effectiveness of LDRMs especially in the low data regime. The results on GLUE are to showcase that SURMs can be extended to other modalities (text) and different modeling regimes (adapters). We will make this more clear in the manuscript.
>
>
> [80] A large-scale study of representation learning with the visual task adaptation benchmark, Zhai et al. 2020.

---

> ### Author Response · Authors · 2024-08-11
>
> > Large data Regime
>
> We investigate the performance of SURM in a large data regime using the iNat2021 [1] dataset. iNat2021 has over **2.7 million** training images, 100K validation images, and 500K test images, representing a wide array of **10,000 species (classes)**.
>
> Full fine-tuning: **69.98%** vs SURM (circulant): **69.01%**
>
> We observe that SURM achieves similar results to full-finetuning using only **55K** parameters as opposed to **86M** in full fine-tuning.. We will add these details to the manuscript.
>
> [1] Benchmarking Representation Learning for Natural World Image Collections, Horn et al. 2021..
>
>  > Large Model regime
>
> Following the suggestion reviewer o41z, we created 10k pairs of UUIDs and tested the memorization capability of LLMs. We use the large-scale **Llama-2-7B** model [2]. The goal of this experiment is to show that models struggle to learn out-of-distribution data when using low-rank updates. Since the pdf can not be updated, the results are presented below:
>
> | Method &#8595; / Steps &#8594;    | 25   | 100  | 200  | 300  | 400  | 500  | 600   | 700   | 800   | 900   | 1000  | 3000 |
> |----------------------------------|------|------|------|------|------|------|-------|-------|-------|-------|-------|------|
> | LoRA (2.9% Param)                | 2.42 | 2.29 | 2.30 | 2.29 | 2.28 | 2.28 | 2.28  | 2.28  | 2.29  | 2.27  | 2.29  | 2.29 |
> | MoRA (2.9% Param)                | 2.43 | 2.29 | 2.30 | 2.28 | 2.28 | 2.28 | 2.29  | 2.28  | 2.29  | 2.27  | 2.29  | 2.26 |
> | Circulant (0.01% Param)          | 3.40 | 3.32 | 2.96 | 2.70 | 2.35 | 2.04 | 1.92  | 1.83  | 1.78  | 1.74  | 1.72  | 0.97 |
> | Circulant+Skew-circulant (0.04%) | 3.78 | 0.34 | 0.09 | 0.05 | 0.02 | 0.02 | 0.006 | 0.009 | 0.004 | 0.003 | 0.001 | 0.0  |
>
>
> We observe that LoRA and MoRA struggle to fit the data (cross-entropy loss around 2.3) whereas our circulant variant achieves a loss of **0.97**. In this experiment, we used a high rank=256 for both LoRA and MoRA, and modified the $Q, K, V$ parameters for all methods.
>
>
> Furthermore, we show the effect of increasing the number of training parameters by using sums of products of circulant and skew-circulant matrices. A skew circulant matrix $S = (s\_{jk})\_{j,k=0}^{n-1}$ is said to be skew-circulant if $s\_{jk} = s\_{j−k}$ and $s\_{−l} = −s\_{n−l}$ for $1 \leq l \leq n − 1$. The motivation for using this particular sum of products comes from the approximation quality of such matrices (see Theorem 1 in [3]). This is evident in practice as circulant+skew-circulant variant obtains a loss=**0** and converges much faster.
>
> This result and our toy experiment (Figure 5) consistently show that low rank updates may struggle to fit various data regimes and that unrestricted-rank matrices may be required to alleviate this issue. SURMs solve this problem using structured matrices (keeping the parameter budget low) while allowing for arbitrary ranks.
>
> We will add this result to the manuscript.
>
> [2] MoRA: High-Rank Updating for Parameter-Efficient Fine-Tuning Jiang et al. 2024
>
> [3] Structured Transforms for Small-Footprint Deep Learning Sindhwani et al. 2015

---

> ### Author Response · Authors · 2024-08-12
>
> Dear Reviewer GJ7U,
>
> We would like to once more sincerely thank you for all the comments and very useful feedback. We think that we have addressed in depth all Reviewer's questions. Please let us know. If the Reviewer has any additional questions, we would be more than happy to answer them.
>
> Yours sincerely,
>
> The Authors

---

> > ### Comment · Reviewer_GJ7U · 2024-08-13
> > **Thanks for Response**
> >
> > Thanks for the authors' response. My main concerns have been addressed. So I will keep my score.

---

### Official Review · Reviewer_aa5x · 2024-07-13

**Soundness:** 3
**Presentation:** 3
**Contribution:** 3
**Rating:** 6
**Confidence:** 2

**Summary:**

This paper explores structured unrestricted-rank matrices (SURM) for parameter-efficient fine-tuning (PEFT) of large-scale Transformer models. This method (SURM) was the first to apply low displacement rank matrices (LDRM) which could support fast matrix-vector multiplication and showed flexibility in finding a balance between compactness and expressiveness. What they used were Circulant, Toeplitz (LDRM), also Kronecker matrix (not LDRM). The authors demonstrated improved accuracy in various data sets while replacing low-rank matrices in LoRA and showed a reduction in the number of parameters in the adapter.

**Strengths:**

1. Using Circulant and Toeplitz for PEFT seems to be a novel technique and also very effective on many standard benchmarks.

2. Extensive experiments have been provided, which support the authors’ claim.

3. The writing is clear, and comprehensive supplementary material would be very helpful for the readers.

**Weaknesses:**

1. Any reasons why the proposed version w/ the Kronecker product outperforms the previous works, such as Kadaptation?

2. Why is PSD approximation accuracy a good proxy task? is the weight delta during finetuning close to PSD by any chance?

3. As the author mentioned in the limitation section, there are many fancy accelerated implementations of LoRA. How slow is the proposed method compared to those LoRA implementations?

**Questions:**

see the weakness section.

**Limitations:**

I do not see any serious societal impact of the paper.

---

> ### Author Rebuttal · Authors · 2024-08-06
>
> We sincerely thank the Reviewer for their valuable feedback and comments.
>
> > Any reasons why the proposed version w/ the Kronecker product outperforms the previous works, such as Kadaptation?
>
> Kadaptation approximates the gradient update using: $\Delta W = \sum_{i=1}^n A_i \bigotimes B_i$ where $B_i$ is the outer product of two vectors (thereby is low rank) and $\bigotimes$ is the Kronecker product. In our case, we consider $n=1$ and do not require $A_i, B_i$ to be low rank. The uplift in accuracy also showcases the need to move away from low rank constraints on the $\Delta W$. These differences are more elaborated in Appendix F.
>
> > Why is PSD approximation accuracy a good proxy task? is the weight delta during finetuning close to PSD by any chance?
>
> The update $\delta W$ weights are close to full rank and thus we used PSD as a proxy for it. Across multiple applications we find that the $\delta W$ after finetuning is full rank. We will add details regarding the rank of the update matrix for general finetuning.
>
> > As the author mentioned in the limitation section, there are many fancy accelerated implementations of LoRA. How slow is the proposed method compared to those LoRA implementations?
>
> Theoretically (please see lines 122-130) we are faster with $O(n \log n)$ whereas LoRa is $O(n d)$. Practically the Circulant variant of SURM is at par with LoRA whereas Toeplitz is slower.

---

> > ### Comment · Reviewer_aa5x · 2024-08-13
> > **full rank vs PSD?**
> >
> > Thanks for your response, I appreciate it.
> >
> > You mentioned $\delta W$ is almost full rank and you used PSD as a proxy for it. Why is a PSD matrix a good proxy for full-rank matrices? Am I missing something?

---

> > > ### Author Response · Authors · 2024-08-13
> > >
> > > Thank you for the question.
> > >
> > > Our goal is to show the approximation capabilities of SURM over a wide class of matrices. We compared SURMs capabilities to full rank random matrices in Fig 4 (top left). The next class we explored is Positive Definite matrices, which have full rank and some structure. It’s worth noting that trained neural network weights also exhibit certain structural patterns [4, 5].
> > >
> > >
> > > In general, (Symmetric) Positive Definite (SPD) matrices are important in various applications in ML like convex optimization and kernel learning.  SPD-NN is widely used in Riemannian optimization [1], manifold learning [2], and CNNs [3] among others. Given the extensive literature on the use of SPD matrices in machine learning, they provide an ideal test case to evaluate the approximation quality of SURM.
> > >
> > > [1] Riemannian Multinomial Logistics Regression for SPD Neural Networks, Chen et al. CVPR 2024
> > >
> > > [2] A Neural Network Based on SPD Manifold Learning for Skeleton-Based Hand Gesture Recognition Nguyen et al. 2019
> > >
> > > [3] U-SPDNet: An SPD manifold learning-based neural network for visual classification Wang et al. 2023
> > >
> > > [4] The training process of many deep networks explores the same low-dimensional manifold, Mao et al. 2024
> > >
> > > [5]  Principles of Riemannian Geometry in Neural Networks, Hauser et al. 2017

---

> ### Author Response · Authors · 2024-08-11
>
> > Large data Regime
>
> We investigate the performance of SURM in a large data regime using the iNat2021 [1] dataset. iNat2021 has over **2.7 million** training images, 100K validation images, and 500K test images, representing a wide array of **10,000 species (classes)**.
>
> Full fine-tuning: **69.98%** vs SURM (circulant): **69.01%**
>
> We observe that SURM achieves similar results to full-finetuning using only **55K** parameters as opposed to **86M** in full fine-tuning. We will add these details to the manuscript.
>
> [1] Benchmarking Representation Learning for Natural World Image Collections, Horn et al. 2021..
>
>  > Large Model regime
>
> Following the suggestion reviewer o41z, we created 10k pairs of UUIDs and tested the memorization capability of LLMs. We use the large-scale **Llama-2-7B** model [2]. The goal of this experiment is to show that models struggle to learn out-of-distribution data when using low-rank updates. Since the pdf can not be updated, the results are presented below:
>
> | Method &#8595; / Steps &#8594;    | 25   | 100  | 200  | 300  | 400  | 500  | 600   | 700   | 800   | 900   | 1000  | 3000 |
> |----------------------------------|------|------|------|------|------|------|-------|-------|-------|-------|-------|------|
> | LoRA (2.9% Param)                | 2.42 | 2.29 | 2.30 | 2.29 | 2.28 | 2.28 | 2.28  | 2.28  | 2.29  | 2.27  | 2.29  | 2.29 |
> | MoRA (2.9% Param)                | 2.43 | 2.29 | 2.30 | 2.28 | 2.28 | 2.28 | 2.29  | 2.28  | 2.29  | 2.27  | 2.29  | 2.26 |
> | Circulant (0.01% Param)          | 3.40 | 3.32 | 2.96 | 2.70 | 2.35 | 2.04 | 1.92  | 1.83  | 1.78  | 1.74  | 1.72  | 0.97 |
> | Circulant+Skew-circulant (0.04%) | 3.78 | 0.34 | 0.09 | 0.05 | 0.02 | 0.02 | 0.006 | 0.009 | 0.004 | 0.003 | 0.001 | 0.0  |
>
>
> We observe that LoRA and MoRA struggle to fit the data (cross-entropy loss around 2.3) whereas our circulant variant achieves a loss of **0.97**. In this experiment, we used a high rank=256 for both LoRA and MoRA, and modified the $Q, K, V$ parameters for all methods.
>
>
> Furthermore, we show the effect of increasing the number of training parameters by using sums of products of circulant and skew-circulant matrices. A skew circulant matrix $S = (s\_{jk})\_{j,k=0}^{n-1}$ is said to be skew-circulant if $s\_{jk} = s\_{j−k}$ and $s\_{−l} = −s\_{n−l}$ for $1 \leq l \leq n − 1$. The motivation for using this particular sum of products comes from the approximation quality of such matrices (see Theorem 1 in [3]). This is evident in practice as circulant+skew-circulant variant obtains a loss=**0** and converges much faster.
>
> This result and our toy experiment (Figure 5) consistently show that low rank updates may struggle to fit various data regimes and that unrestricted-rank matrices may be required to alleviate this issue. SURMs solve this problem using structured matrices (keeping the parameter budget low) while allowing for arbitrary ranks.
>
> We will add this result to the manuscript.
>
> [2] MoRA: High-Rank Updating for Parameter-Efficient Fine-Tuning Jiang et al. 2024
>
> [3] Structured Transforms for Small-Footprint Deep Learning Sindhwani et al. 2015

---

> ### Author Response · Authors · 2024-08-12
>
> Dear Reviewer aa5x,
>
> We would like to once more sincerely thank you for all the comments and very useful feedback. We think that we have addressed in depth all Reviewer's questions. Please let us know. If the Reviewer has any additional questions, we would be more than happy to answer them.
>
> Yours sincerely,
>
> The Authors

---

### Decision · Program_Chairs · 2024-09-25

**Decision:**

Accept (poster)

**Comment:**

The paper proposes a general framework for parameter-efficient fine-tuning, based on structured unrestricted-rank matrices (SURMs) to substitute LoRA or other parameter-efficient finetuning methods.

Three variants of SURMs are included, named Kronecker, Toeplitz, and Circulant, based on the matrix type. The use of Circulant and Toeplitz structures in adaptors is novel.

The extensive experiments show that the SURM method generally achieves performance comparable with LoRA/adaptors with 50% or less parameters on vision datasets like CIFAR10, SUN397, and DTD. SURM can also be incorporated into adaptors.

The rebuttal includes results on larger LLM models such as LLaMa-2 7B, where existing methods such as LoRA struggle, while SURM does better, by allowing higher ranks without higher computational load.

The bibliography is poorly constructed and needs to be fixed. Many of the entries are missing information, such as journal or conference name, or even arxiv numbers.

The authors will have a challenge to fit in all the new material (e.g. the results on the LLaMa experiments) into the final version if accepted.

The low-rank adaptors and approximations space is becoming very crowded and hard for new methods to stand out as significant contributions. This is the case for this paper. While the contribution is clear, and makes a good addition to the current PEFT methods, it might not stand out as particularly exciting in comparison with other accepted papers. Hence it is suited more for a poster or spotlight than an oral talk.